# Exploring the Relationship between Micronutrients and Athletic Performance: A Comprehensive Scientific Systematic Review of the Literature in Sports Medicine

**DOI:** 10.3390/sports11060109

**Published:** 2023-05-24

**Authors:** Hadeel Ali Ghazzawi, Mariam Ali Hussain, Khadija Majdy Raziq, Khawla Khaled Alsendi, Reem Osama Alaamer, Manar Jaradat, Sondos Alobaidi, Raghad Al Aqili, Khaled Trabelsi, Haitham Jahrami

**Affiliations:** 1Department Nutrition and Food Technology, School of Agriculture, The University of Jordan, Amman 11942, Jordan; manar_jaradat98@outlook.com (M.J.); sondosalobaidi@gmail.com (S.A.);; 2Department of Psychiatry, College of Medicine and Medical Sciences, Arabian Gulf University, Manama 323, Bahrain; mariamahh@agu.edu.bh (M.A.H.); khadijamz@agu.edu.bh (K.M.R.); khawlaalsendy948@gmail.com (K.K.A.);; 3High Institute of Sport and Physical Education of Sfax, University of Sfax, Sfax 3000, Tunisia; 4Research Laboratory—Education, Motricity, Sport and Health, University of Sfax, Sfax 3000, Tunisia; 5Government Hospitals, Ministry of Health, Manama 323, Bahrain

**Keywords:** vitamins, minerals, sport, performance

## Abstract

The aim of this systematic review is twofold: (i) to examine the effects of micronutrient intake on athletic performance and (ii) to determine the specific micronutrients, such as vitamins, minerals, and antioxidants, that offer the most significant enhancements in terms of athletic performance, with the goal of providing guidance to athletes and coaches in optimizing their nutritional strategies. The study conducted a systematic search of electronic databases (i.e., PubMed, Web of Science, Scopus) using keywords pertaining to micronutrients, athletic performance, and exercise. The search involved particular criteria of studies published in English between 1950 and 2023. The findings suggest that vitamins and minerals are crucial for an athlete’s health and physical performance, and no single micronutrient is more important than others. Micronutrients are necessary for optimal metabolic body’s functions such as energy production, muscle growth, and recovery, which are all important for sport performance. Meeting the daily intake requirement of micronutrients is essential for athletes, and while a balanced diet that includes healthy lean protein sources, whole grains, fruits, and vegetables is generally sufficient, athletes who are unable to meet their micronutrient needs due to malabsorption or specific deficiencies may benefit from taking multivitamin supplements. However, athletes should only take micronutrient supplements with the consultation of a specialized physician or nutritionist and avoid taking them without confirming a deficiency.

## 1. Introduction

Optimal performance is a primary objective for many athletes, and this can be attained by following a suitable exercise protocol and ensuring proper nutrition [1]. Food is composed of nutrients that are essential for growth, repair, and energy generation depending on the amount that meets the body’s need [2]. Nutrients are typically categorized into two main groups: micronutrients and macronutrients [3]. When it comes to athletic performance, the importance of micronutrients should not be neglected [4]. Athletes are prone to consuming insufficient amounts of micronutrients due to inappropriate dietary habits, especially if they are not matching their physical activity requirements [5]. By making sure they are receiving adequate levels of micronutrients, athletes can give themselves a competitive edge and maximize the potential of their training [6]. Micronutrients may boost mental performance, help balance hormones, and keep cognitive performance at its peak [7].

It is noteworthy to emphasize that sports nutrition is not a one-size-fits-all solution, as each athlete has specific nutritional needs [7]. Therefore, athletes, nutritionists, and coaches must work together to customize nutritional plans for each athlete to ensure that their athletes/players’ needs are considered properly and they are receiving the sufficient level of nutrients they need to aid in the adaptation to their training and ultimately support optimal athletic performance. The evaluation of the evidence of the impact of micronutrients on the performance of athletes is the main purpose of this comprehensive systematic review paper.

## 2. Methods and Search Strategy

A comprehensive literature search was conducted using the Web of Science, Scopus, and PubMed databases. In order to retrieve relevant studies on the topic, our search strategy incorporated keywords including “micronutrients”, “vitamins”, “minerals”,” antioxidants”, “athletes”, “sport performance”, “training”, and “exercise”. Original research articles involving human subjects, English-language publications, human subjects, and a focus on micronutrients and athletic performance were the inclusion criteria. For the systematic review procedure, the Synthesis without meta-analysis (SWIM) recommendations were followed. The search was limited to articles published between January 1950 and 31 March 2023. The studies that made the cut for the review had to measure outcomes related to athletic performance, have a sample size of at least 10 participants, and use an intervention involving a micronutrient supplement. Studies that concentrated on macronutrients—such as carbohydrates and protein—were disregarded.

### 2.1. Data Extraction

Studies’ titles and abstracts located by the search were examined by two independent reviewers. After that, full-text articles were examined to see if they qualified for inclusion in the review. Data on the study design, sample size, intervention protocol, micronutrient supplements used, athletic performance outcomes assessed, and outcomes were extracted from the eligible studies.

### 2.2. Data Synthesis

To summarize the conclusions of the included studies, a narrative synthesis was carried out. The studies’ findings were categorized by micronutrient supplement and athletic performance outcome measures. A description of each study’s design, sample size, intervention strategy, and findings was included in the synthesis. It is important to ensure the accuracy and reliability of a systematic review by ensuring that all information is extracted in a standardized and consistent manner. In this study, two authors independently extracted all information from each paper to minimize the risk of bias and errors. This approach helps to ensure the validity of the review’s findings and strengthens the overall quality of the study. By having two authors independently extract information from each paper, the review can ensure data accuracy and increase confidence in the conclusions drawn from the analysis. It is a rigorous method that is commonly used in systematic reviews and emphasizes the importance of transparency and objectivity in research [8].

## 3. Results

A total of 231 articles were obtained involving 18,683 athletes. Table 1 provides a summary of the main micronutrients researched in sports medicine. Table 2 provides a summary of all available 217 research articles about micronutrients and sports performance. Figure 1 simplified the model of micronutrients’ main functions in sports medicine.

## 4. Vitamins

Vitamins are organic essential compounds that cannot be synthesized by the human body [29]. They play a vital role in numerous functions that are relevant to the athlete’s performance [7]. Their functions are evident in co-enzymes, hormones, and autoxidation, as well as their contribution to energy production [242]. There are thirteen various kinds of vitamins currently divided into two major groups due to their chemical and biological functions, four of which are fat-soluble vitamins (FSV) including vitamins A, D, E, and K, and the rest are water-soluble vitamins (WSV) including B complex vitamins and vitamin C [243]. Table 1 summarizes the recommendation requirements of vitamins along with the rich sources and their roles in exercise performance.

## 5. Fat-Soluble Vitamins (FSV)

### Vitamin A

Vitamin A plays a significant role in promoting the overall wellbeing of athletes, as it aids in the formation of healthy tissues and improves oxygen access throughout the body, thereby supporting the maintenance of an adequate level of physical activity [3]. It also has a crucial influence on vision, skin health, and immune system functioning [29]. Moreover, vitamin A is a potent antioxidant that helps in neutralizing free radicals generated by oxidative stress during advanced physical training. The sufficient consumption of vitamin A may help alleviate the reactive oxygen species and avoid the onset of illnesses such as heart failure and muscle damage [244], as mentioned in many studies in Table 2.

Vitamin A mainly exists in two forms: animal source (retinol) and plant-based provitamin A (carotenoids). The intake of sufficient amounts of beef liver, eggs, dairy products, and seafood as well as dark leafy green vegetables may ensure meeting the dietary requirements for athletes. It is noteworthy that athletes may benefit from supplementation with multivitamins that contain vitamin A, as a part rather than its own sole use [29].

Vitamin A has been proved to enhance and support various body functions, including reaction time, muscles recovery [245,246], and protein synthesis, which is essential for muscle growth and recovery and can be important for those competing in events requiring fast reflexes [247]. Furthermore, vitamin A can help protect athletes against injuries by increasing healing times and promoting the formation of healthy connective tissues [3]. Finally, vitamin A may help fight off colds, flu, and other illnesses, which can be particularly helpful for traveling athletes.

To evaluate the effect of crocetin on fatigue, a study test was conducted by athletes to measure stamina using a bicycle ergometer at a standard workload for 120 min twice. They also performed non-workload tests of 10 s at 30 min at a maximum velocity (MV) [242]. A difference in MV from 30 to the 210 min test was remarkably reported only in men who consumed crocetin when compared to their counterpart who used a placebo (*p* < 0.05) [248]. No difference was observed with the consumption of ascorbic acid in all candidates with the same period [248]. The daily consumption of crocetin would reduce physical exhaustion in men, according to these results [179]. The attenuating effect of saffron carotenoids on muscle fatigue is due to their provitamin A activity [248]. Athletes should meet their daily recommended intake of vitamin A to guarantee a perfect peak physical condition [249].

The Recommended Dietary Allowance (RDA) of retinol activity equivalents (mcg RAE), as shown in Table 1, is 900 micrograms for men aged 19 years old and older, equal to 3000 international units (IU), and 700 mcg RAE for women aged 19 years old and older, equivalent to 2333 (IU). However, the adherence to a maximum Tolerable Upper Limit (UL) of 10,000 IU (3000 mcg)/day for adults is important to avoid any dangerous effect. It is also important to emphasize that toxicity may occur when frequent doses of more than 25,000 IU are taken daily [3].

β-carotene is a member of the carotenoid family that is thought to provide numerous health benefits, including immunity system boosting, antioxidation properties, and performance enhancement [250]. β-carotene is an essential antioxidant, meaning that it helps prevent the harmful effect of free radicals on cells [251]. This is important for athletes, as free radicals are generated during strenuous exercise and can lead to fatigue and soreness [176]. Several studies have reported similar findings, highlighting the ability of vitamin A to potentially reduce recovery time from exercise [7,250]

Sumac juice drink was tested in a study to evaluate its impact on pain scores through post-exercise intervals. Forty healthy candidates involved in an aerobic training protocol for four weeks received a dose of placebo or sumac juice consumption two times/day for a month. The results revealed that participants of the sumac juice group had a lower pain score increment and even a better enhancement during pain relief. The potency of protecting muscles might be due to the sumac juice antioxidant potency of β-carotene-linoleic acid components. These results highlight the possibility of sumac juice consumption in improving muscle performance among athletes [252]. Nevertheless, future studies of athletes are warranted.

Athletes often push their bodies to the limit, therefore compromising their immune systems and making them more susceptible to infections and illness [176]. Taking a supplement with β-carotene may have potential benefits for the immune system, helping to prevent illness and potentially enabling athletes to train harder and longer [250,251]. Apparently healthy nonsmoker adult males were involved in consuming placebo or 15 mg/day of β-carotene for 26 days. After oral administration, significant increases in the monocytes percentage representing the major histocompatibility complex class II molecule human leukocyte antigen DR isotype (HLA-DR) and the adhesion molecules intercellular adhesion molecule-1 and leukocyte function-associated antigen-3 were observed. Furthermore, tumor necrosis factor-alpha (TNF-alpha) was notably elevated due to the dose intake, arguing that a slight increment in the consumption of dietary β-carotene can enhance the responses of immune cells within the short term, supporting the process of the carcinogenic potency [253].

Lastly, β-carotene has been found to improve physical performance [254]. Studies have shown that β-carotene supplementation may lead to endurance strength improvement and injury risk prevention [3,150]. Similarly, a meta-analysis of nine studies including participants from both genders, with a total 190,545 candidates, emphasized that β-carotene leads to a significant enhancement in overall performance [29]. There was a 95% possibility that β-carotene consumption attenuates the possibility of hip fracture and other different fracture types by over 20% [251]. According to research, despite the uncertain effect of the antioxidant’s supplementation, consuming β-carotene and combining antioxidants or not still has an effective impact in reducing exercise-induced peroxidation [33]. This may enhance athletes’ endurance in the long term [251,254,255].

## 6. Vitamin E

As research progresses, the potential advantages of vitamin E for athletes are becoming increasingly apparent. Vitamin E helps in protecting athletes’ bodies and may improve both performance and recovery [75]. Overtraining and intense exercise are associated with reactive oxygen species (ROSs) production, which aids in enhanced muscular and endurance adaptation to exercise through the upregulation of endogenous antioxidant enzymes [256]. However, excess accumulation of ROS accompanied by the inability of the body to scavenge these compounds is harmful to the body’s cell components which is associated with fatigue, delayed recovery, and reduced performance [257]. Accordingly, research suggests the possible protective effects of vitamin E supplements against chronic stress associated with exercise [257,258]. This vitamin possesses antioxidant properties by neutralizing free radicals, protecting cells and tissues [257,258].

Studies showed controversial results of vitamin E supplementation in athletes [44]. It was shown that vitamin E supplements neither inhibit exercise-induced oxidative stress nor impact endurance running performance [259]. Moreover, these results were supported by the randomized controlled trial on athletic students where vitamin E oral consumption was found to not influence exercise endurance [260]. On the other hand, among healthy individuals, vitamin E (alpha-tocopherol) consumption inhibits the exercise-induced reduction in blood paraoxonase 1/arylesterase activity [6,44,147,260]. Excess doses of supplements have been shown in studies to inhibit the signaling reactions required for adaptations to exercise, creating an interference effect [3].

A meta-analysis revealed that vitamin E supplements have a beneficial and protective effect, particularly at low doses (≤500 IU/day), in reducing biomarkers associated with exercise-induced muscle damage and oxidative stress. Beneficial effects of the antioxidant characteristics of vitamin E were observed among exercise-induced side-effects [257]; both animal and human studies have indicated that Vitamin E has the potential to enhance immune function and provide protection against various infectious diseases [6]. Vitamin E reduces PGE2 production and inhibits COX2 activity, likely by decreasing nitric oxide production [261]. Furthermore, it improves T cells immune synapse formation and activation signals, and lastly, it helps in modulating the T (Th1/Th2) balance [52]. This is particularly beneficial for athletes, who are often susceptible to illness and injury due to the intense physical demands of their sport. By supplementing with vitamin E, athletes can help increase the body’s natural immunity, thus reducing the risk of infection and promoting recovery [262].

Vitamin E may assist in improving blood flow, which is essential for athletes [242]. γ-tocopherol, which is one of the compounds that contain vitamin E, increases cardiovascular functions. γ-tocopherol expands the activity of nitric oxide synthase, which in turn produces nitric oxide, aiding in vessels relaxation and thus improving blood flow [29]. Adequate levels of nutrients and oxygen concentrations in muscles indicate a raised blood flow rate, which may help improve performance [263]. Additionally, vitamin E improves red blood cells’ flow and flexibility [264]. This is important for athletes, as improved blood flow means better performance in delivering nutrients and oxygen to the muscles, allowing them to perform at their best [176].

Free radicals such as superoxide, nitric oxide, and hydrogen peroxide are known to be of significant importance, as there must be a balance between antioxidants and free radicals in order to obtain physiological muscle adaptation in response to exercise [25]. Few studies suggested that antioxidants supplementation may be beneficial under specific circumstances, such as overtraining, high-altitude training, or hypoxic training, and claimed that antioxidant usage such as the intake of vitamin E or vitamin C may have no benefit at all or may even cause harm [6,7,29,176,256]. Misusing or consuming excessive amounts of vitamins can lead to muscle fatigue and impede the recovery process due to the inactivation of the gene expression regulator Nrf2 (Nuclear factor erythroid 2-related factor 2), which plays a role in the response to cellular stress and contributes to enhancing exercise performance [265]. It is worth noting that Vitamin E toxicity may cause increases in mortality risk factors, since there has been a positive relation accompanied by a high-sensitivity C reactive protein indicative of inflammation [266].

## 7. Vitamin D

Vitamin D plays a cooperative role in the synthesis of various hormones in the body [3]. Dairy products, egg yolk, and fatty fish are the rich dietary intake sources [2]. Moreover, it can be synthesized in vivo and be activated by sunlight within a duration of 15 to 20 min of exposure [44]. It also plays an important role in calcium homeostasis and constant healthy bone [30], functions of improving the immune system, musculoskeletal system, power, and force output [45].

Vitamin D supplementation has been increased among athletes [45]. Unfortunately, the widespread vitamin D insufficiency has been clearly stated in elite male athletes, with evidence of a deficit in women [46]. Percentages of insufficiency in elite athletes were above 50%, and the deficiency in other studies was 70–90%, as reported by Harju et al. [47]. Certain circumstances impact vitamin D status, such as indoor training, pigmented skin, and living in a high-altitude region [48]. Studies have reported that athletes with vitamin D deficiency may experience ergogenic benefits when taking vitamin D supplements [103].

There was a direct relation between the concentrations of vitamin D and athletes’ performance, such as speed, jumps’ height, power muscle tone, and strength of handgrip [50]. Moreover, the addition of calcium to vitamin D supplements exhibited a reduction in the stress fracture rate [106,138].

In a study conducted among 70 athletes randomly assigned for 8 weeks to either vitamin D oral supplements of 50,000 IU/week group or a control group, a significant improvement in the test of the strength leg press in both groups was reported [267]. However, the results emphasized that the enhancement in the supplemented group was obviously more noted than that in the control group (*p* = 0.034). Moreover, when the sprint test was conducted, within-group enhancement had been noticed in the supplemented group only (*p* = 0.030). The results showed that regular weekly vitamin D supplementation with a dose of 50,000 IU increased the levels of circulating calcidiol (major circulating form of vitamin D) by approximately 17 ng/mL. This increment was related to a notable enhancement in sprint and power leg examinations in the vitamin D group [138]

Additionally, vitamin D is also thought to improve the body’s utilization of carbohydrates during exercise, providing the body with increased energy, which can help to enhance performance [1]. Twenty-two male adult athletes were allocated into two groups for 8 days: a one-shot dose of 150,000 IU vitamin D group and a placebo group. The vitamin D group showed a significant elevation in muscle power in the period from day 1 up to day 8, suggesting that a single dose of 150 000 IU vitamin D had a beneficial impact on serum 25-hydroxy vitamin D (25(OH)D) levels and the muscle’s role [268].

To maintain sufficient vitamin D levels, the most appropriate way is to spend time in outdoor direct sunlight for several minutes each day, as obviously indicated by many studies’ conclusions (see Table 2). This prescription for sun exposure should also be combined with foods that are rich in vitamin D, such as dairy products, fatty fish, and fortified foods [101]. Additionally, athletes may also benefit from daily multivitamin supplements that contain vitamin D to ensure that their body receives the best possible nutrition. The recommended daily dose of vitamin D, as shown in Table 1, varies depending on age; a daily dose of 600 IU (equivalent to 15 micrograms (mcg)) is considered sufficient for the age of 19 and above in both genders, and for adults over 70 years old, an 800 IU (20 mcg) daily dose would be sufficient [7,25]. Vitamin D-deficient athletes would require 50,000 IU of vitamin D per week for 8 weeks [138,258].

A previous study was conducted for 12 weeks among 53 youth athlete swimmers who suffered from insufficient levels of vitamin D to evaluate the influence of vitamin D oral supplementation on physical performance by taking 2000 IU/day of vitamin D or placebo. No notable difference was observed in performance between the supplemented and placebo groups [269]. The results concluded that there was no remarkable correlation found between Vitamin D levels and the evaluated criteria including strength or swimming performance and even the age-adjusted balance. Although the oral administration of vitamin D had raised the concentration of Vitamin D compared to the placebo group, no significant physical performance enhancement was reported [269].

## 8. Vitamin K

Vitamin K is essential for blood coagulation [163]. It may also impact bone metabolism in postmenopausal women, according to a few previous studies [3,7,270]. In elite female athletes, the oral intake of vitamin K at a dose of 10 mg/day has been shown to improve bone remodeling [254] by increasing the calcium-binding capacity of osteocalcin, promoting bone formation, and reducing bone resorption [176]. Moreover, the intake of vitamin K improved cardiovascular function [18,231]. Table 1 summarized the recommendation and the role of vitamin K in exercise performance.

## 9. Water-Soluble Vitamins (WSV)

### Vitamin B

B-complex vitamins are essential for athletes to maintain optimal health and performance [19]. B-complex vitamins help athletes manage stress and anxiety, aid in muscle recovery, and reduce fatigue, which may adversely affect performance if left unchecked [2]. B-complex vitamins help in blood pressure regulation [271]. Moreover, B-complex vitamins aid in maintaining a healthy sleep schedule by regulating levels of the sleep-regulating hormone melatonin, helping athletes fall in a deep continuous sleep [5]. This is essential for athletes, as the lack of sleep can affect an athlete’s performance [52]. B-complex vitamins also contribute to maintaining optimal health and performance in athletes, supporting improved brain functioning, concentration, sleep quality, and energy levels [19]. Thus, athletes need to ensure that they are receiving enough vitamin B through their diet or supplements [244].

***Thiamine (B1)*** is a water-soluble vitamin that must be consumed regularly from the diet [6]. Although free thiamine is stable at acidic pH, it is destroyed by ultraviolet (UV) and gamma irradiation and is heat-sensitive [29]. Whole grains, bread, and nuts are the most common thiamine food sources, while milled wheat flour, polished rice, vegetables, and fruits contain less thiamin [272]. The large intestine’s bacteria in the human body are able to produce thiamine and thiamine pyrophosphate (TPP) [52]. Thiamine leaches into the water due to its solubility and is inevitably lost in any discarded soaking or cooking water, as well as destroyed by heating during culinary methods [7].

Thiamine, in its active state (TPP), is a cofactor of numerous important enzymes involved in the metabolism of carbohydrates and branched-chain amino acids [7]. Moreover, it is necessary for several other cellular functions, including the development of nucleic acid precursors, myelin, and neurotransmitters (such as acetylcholine), as well as antioxidant defense [272]. A deficiency of this vitamin leads to a decline in oxidative metabolism [265]. The biochemical outcomes include a failure to create adenosine triphosphate (ATP), lactic acidosis resulting in a greater lactic acid generation, and a reduction in neurotransmitter synthesis (e.g., acetylcholine, glutamate, aspartate, and gamma-aminobutyric acid (GABA)) [6]. The major causes of thiamin deficiency are either the insufficient intake, poor absorption or metabolism, or an increase the body demand [265]. Furthermore, diuretics and diarrhea lead to thiamine deficiency [272]. Regarding thiamin and exercise, research suggests that thiamin availability in the diet appears to influence exercise capacity when athletes consume the recommended amount [254].

***Riboflavin (B2)*** is the second vitamin from the B-complex vitamins [270]. It appears as a yellow-orange chemical molecule that is water-soluble [273]. Riboflavin is relatively heat- and oxygen-stable, especially in an acidic environment [19]. It is very light-sensitive, destroyed by reducing agents, and unstable in alkaline solutions [4]. Riboflavin is essential and must be obtained from food sources [2]. Riboflavin is abundant in almonds, beef liver, sardines, mushrooms, cheddar cheese, and eggs [271]. When athletes consume a typical amount of riboflavin, their exercise capability would be optimum [254].

***Niacin (B3)*** is the third water-soluble member of the B vitamins family [176]. Humans can partially convert the essential amino acid tryptophan to nicotinamide, which is a dietary supply of niacin [270]. However, the conversion cannot meet the demands for niacin, so dietary niacin supplies around 50% of the daily niacin requirement [256]. Meat, whole grains, milk, and dairy products are good sources of niacin [242]. Niacin is abundant in peanuts, seafood, mushrooms, and yeasts [29]. Food items high in tryptophan-containing proteins, such as milk, cheese, and eggs, are good sources of niacin [7]. Its roles include reduction and oxidation (redox) processes, as well as acting as a ligand for a range of purine receptors [243].

It is hypothesized that this vitamin lowers cholesterol, improves thermoregulation, and improves oxidative metabolism [274]. In hypercholesteremic individuals, a niacin intake of 100–500 mg/day may help lower blood lipid levels while increasing homocysteine levels [176]. Nevertheless, consuming 280 mg of niacin during exercise has been demonstrated to reduce exercise capacity by moderating fatty acid mobilization [275].

***Pantothenic acid (B5)*** is a water-soluble vitamin that is widely available in the diet [3]. It is often provided as calcium pantothenate, which is more stable against light, heat, and oxygen, but is unstable in both alkaline and acidic circumstances [273]. Sodium pantothenate is also available, but its use is restricted due to its hygroscopicity [272]. Pantothenic acid functions as a coenzyme for acetyl coenzyme A (acetyl CoA), implying its importance in aerobic or oxygen-based energy systems [2]. Acetyl CoA supplementation has not been shown to increase aerobic performance in studies [25,52,202,276]. Yet, one study found a reduction in the lactic acid buildup, but no benefit in performance was concluded [277].

***Pyridoxine (B6)*** is marketed as a supplement that increases muscular growth, strength, and aerobic capacity in the lactic acid and oxygen systems [141]. It might additionally have a relaxing effect, which has been related to increased mental power [24]. Surprisingly, research showed that pyridoxine did not increase the capacity of aerobic exercise or the accumulation of lactic acid in well-nourished athletes [23]. However, when paired with vitamins B1 and B12, it has been shown to raise serotonin levels and enhance motor abilities, which are required in sports such as pistol shooting and archery [24]. Moreover, vitamin B6, thiamin, and pantothenic acid showed inverse relationships with stress risk and anxiety [25]. Another study revealed that after a month of vitamin B6 intake, young adult athletes reported feeling less anxiety [25]. Table 2 presents studies that investigated the effect of vitamin B6 intake either as part of a multivitamin supplement or as a sole intake on exercise performance. Most of the studies reported a positive impact on exercise performance, particularly in cases of vitamin B6 deficiency.

***Cyano-cobalamin (B1**2)*** is a coenzyme required for the synthesis of deoxyribonucleic acid (DNA) and serotonin [176]. In theory, it would enhance muscular mass and blood oxygen-carrying capacity and lessen anxiety [273]. However, no ergogenic impact has been documented in well-nourished athletes [242]. Interestingly, it may enhance pistol shooting performance due to the stimulation of serotonin production, which reduces anxiety [2]. A cross-sectional research work studied 100 amnestic mild cognitive impairment (MCI) patients characterized by low-normal and high-normal vitamin B12 levels, who were then enrolled in an Auditory Verbal Learning test to evaluate their memory’s function. The results showed that those with low-normal B12 concentrations had notable defects in learning and recognition abilities and even in memory performance due to the low microstructure integrity of the hippocampus [23]. It is important to acknowledge that vitamin B12 is crucial for proper brain functioning, as it aids in faster information processing and enhances concentration levels [21]. This has been demonstrated in patients with mild cognitive impairment (MCI) who had low-normal levels of vitamin B12 [262]. This is especially important for athletes, as improved brain functioning may help improve performance in many ways, from learning new techniques to continuous focus maintenance during long competitions [24].

***Folic acid (folate)*** is a coenzyme that aids in the synthesis of DNA and red blood cells [278]. An increased red blood cell count enhances oxygen supply to muscles during exercise [21,79]. It is thought to be crucial in preventing birth abnormalities and may lower homocysteine levels, which is a risk factor for heart disease [22]. Folic acid supplements did not increase exercise performance among malnourished athletes with folate deficiency [254].

## 10. Vitamin C

Vitamin C (also known as ascorbic acid) can be found in many types of food, including oranges, strawberries, broccoli, and sweet potatoes [15]. Athletes require more vitamin C than the average person since their bodies are working harder and being pushed to the limits [15,279]. Therefore, they need to receive enough of this vital nutrient to perform at their best. Researchers have reported that the intake of vitamin C supplements does not boost physical performance in well-nourished athletes [52]. Nevertheless, athletes are recommended to receive an adequate amount of vitamin C from their balanced diet.

The crucial role of vitamin C in neutralizing free radicals has been raised from its antioxidative potency [69], thereby improving the immune system [15] and reducing the risk of illnesses such as colds and other viruses [280]. It plays an important role in immunity by enhancing the differentiation and proliferation of B and T lymphocytes and increasing antibodies levels [25]. Furthermore, vitamin C has been reported to modulate cytokine production and decrease histamine levels [169,281]. Studies have also shown that vitamin C can eliminate fatigue, improve coordination, and increase endurance [15].

Vitamin C has a crucial role to play in wound healing and collagen production [3]. It helps boost energy levels and protects the body from illnesses and injuries [242]. Vitamin C works as a co-factor to produce collagen for the propyl and lysol hydroxyls enzymes, which stabilize the structure of collagen [29]. Furthermore, vitamin C also enhances collagen gene expression in fibroblasts [7], contributing to the strength and integrity of joints and muscles, which is essential for the success of any athlete. This is supported by the fact that vitamin C is crucial in protecting against ROS damage, enhancing keratinocyte differentiation, lipid synthesis, fibroblast proliferation, and migration, which has been seen to shorten the time of wound healing [28,282].

However, high levels of vitamin C can, in turn, act as a pro-oxidant rather than an antioxidant [29]. The overconsumption of vitamin C supplement decreases exercise-induced adaptation, delays post-exercise recovery, increases lipid peroxidation, and diminishes mitochondrial biogenesis [127]. These effects can hinder skeletal muscle adaptation to exercise [3].

## 11. Minerals

Numerous physiological and metabolic processes in the human body involve minerals [283]. Minerals have physiological effects on the body during exercise, including maintaining a normal heartbeat, oxygen transportation, antioxidation activity, healthy bone, and immune system enhancement [283]. Sufficient levels of minerals are required for optimal performance because many of these processes are enhanced during sports activity [284]. For athletes to perform at their best, maintaining a healthy body necessitates the intake of a variety of nutrients. Some minerals make weightlifting more effective by enhancing athletic performance; these are the minerals that degrade faster when used in sports endeavors [285] and thus need to be replaced routinely among athletes to sustain their performance. Table 1 summarized the recommendation requirements of minerals along with the rich sources and their roles in exercise performance.

## 12. Iron

Iron (Fe) is a crucial mineral for physical performance, and its importance cannot be overstated [286]. When it comes to peak performance, an adequate intake of iron can make all the difference [149]. It helps the body produce red blood cells, which are necessary for transporting oxygen to the muscles [71]. Without enough iron, athletes and other physically active individuals may suffer from fatigue and lethargy as the body struggles to meet the increased demands [71]. A huge part of the pool of plasma iron (almost 80%) is utilized by the bone marrow; this is equivalent to a 20–30 mg/day dose to ensure the efficient production of erythrocyte [31].

In addition to red blood cells production, iron is also important for energy metabolism [31]. It is necessary for converting food into energy, and it helps to ensure that the body can use energy efficiently for physical activities [280]. Iron also helps the body in regulating its temperature, making it an essential nutrient for athletes competing in warm climates or hot weather [149]. Finally, it is important for other bodily functions, such as the immune system, growth, and hormone production [284]. When considering physical performance, it is important to ensure that iron intake is adequate and balanced. The human physiological mechanism preserves the maximum iron [29]. Based on the total compulsory iron depletion that occurs daily and the average of 10% absorption and bioavailability, the World Health Organization (WHO) and other national institutes have estimated iron-recommended doses depending on several characteristics including gender, age, and race. The recommended dietary intake for females is 18 mg, and for males, it is 8 mg [31]. Poorly planned diets, coupled with inadequate levels of exercise, can lead to anemia and other problems associated with low iron levels [286]. This may cause fatigue, poor performance, and a decreased ability to perform physical activities [285]. Therefore, it is necessary to consume a high-quality variant diet that involves iron-rich sources [284].

It is also important to make sure that athletes have enough time to rest and recover between workouts. Iron helps to replenish energy stores and reduce fatigue, so it is important to give the body time to absorb the nutrient [287]. Additionally, certain supplements may also help in providing additional iron to meet the demands of physical performance [205]. Athletic training can result in alterations including higher vascularization (creation of new blood vessels), elevated hematocrit, and higher erythrocyte awareness in the blood, which may lead to an increase in iron needs [122]. A shortage of iron may result from hemorrhages, gastric blood loss, and/or urinary tract bleeding, especially among high-intensity sports [121]. Professionals are predicted to have 70% higher iron needs than non-professionals [287]. Iron deficiency anemia can impede progress in an athlete’s training by reducing oxygen delivery [283]. Lastly, most research concluded that iron supplements do not enhance aerobic performance, unless there is a specific depletion and/or anemia reported [149].

## 13. Calcium

Athletes must be in peak physical condition to perform at their best and make sure their diets are balanced, which is an important part of their training regimen [288]. Calcium (Ca) is among the many nutrients that athletes need to remain healthy [289]. It not only helps to keep bones and muscles strong, but it has also been linked to improved performance in athletes [32]. However, insufficient Ca consumption and elevated Ca depletion may expose a person to osteoporosis [194]. Athletes should make sure to consume an adequate amount of Ca each day as part of their balanced diet, which would achieve around 1500 mg/d [32,188]. The optimal Ca requirement is 1200 mg/day for adolescents and youth, 1000 mg/day for females aged 25 to 50 years old, and 1500 mg/day for postmenopausal females who are not treated with estrogen replacement therapy [33]. With the right diet and exercise routine, they may capitalize on the benefits that Ca has to offer and maximize their performance [32,188].

Numerous studies have indicated that the adequate and consistent consumption of Ca can potentially enhance physical performance in athletes [289], as it plays a crucial role in maintaining muscle strength, which is a key element for exercise performance [35,289]. Additionally, it may help reduce injuries and improve recovery time [32]. It is also known that Ca may protect the bones and joints from stress caused by continuous physical activity [32]. Conversely, improving Ca status with 2000 mg of Ca supplementation has been shown to reduce the risk of developing a stress fracture [289]. Calcium also helps to convert carbohydrates and fat into energy, which can contribute to performance improvement [78,288]. It also helps in reducing fatigue and delaying the onset of muscle soreness [290].

Calcium can be found in many common foods including milk, yogurt, cheese, and dark leafy greens. Other sources include tofu, nuts, fish, and fortified cereals [289]. Additionally, athletes may consider Ca supplements if they are unable to receive the recommended daily intake from their diet. It is important to note that the amount of Ca an athlete needs daily may vary depending on their weight and activity level [289]. Skeletal muscles’ ability to contract and relax depends in part on Ca [291]. The importance of it binding to troponin C for the contraction of muscles has the potential to influence performance [33]. While it is true that training leads to higher Ca loss, primarily through perspiration, the foundations of bone mineralization are Ca, vitamin D, and physical activity [292]. However, in rare circumstances, especially if the diet is low in its nutrient density, physical activity might endanger bones [65]. Every athlete should place a high priority on developing and maintaining optimal bone health, since vigorous physical activity increases the stress fractures risk [85,188].

## 14. Potassium

When it comes to athletes’ health and performance, one mineral that is essential to success is potassium (K) [293]. It is a required nutrient for human health and is necessary for many physiological processes [293]. Adults should not exceed the consumption of 2000 mg sodium/day (Na) or 5 g of salt and have a minimum dose of 3510 mg potassium/day, according to new guidelines established by the WHO [14,281]. It has a crucial role in muscle contractions and helps the body regulate fluid balance, blood pressure, and the heart rate [281]. In addition to its role in muscle contractions, it is also involved in nerve functions and proper electrolyte balance [59], which may be beneficial to athletes who may be sweating during a long practice or game [29]. Furthermore, proper potassium levels can help prevent injuries and help athletes maintain their energy levels [177,230].

Potassium is a great source of energy for athletes [293]. It helps to reduce the amount of lactic acid stored in the muscles [59], which may lead to fatigue as well as maintain a healthy metabolism [35]. It is also involved in the breakdown of carbohydrates, which helps keep energy levels high during intense physical activity [177]. It is unknown if potassium supplementation reduces the occurrence of muscular cramping in athletes. It should be acknowledged that there have been no reports of ergogenic effects [58].

## 15. Magnesium

Magnesium (Mg) is an essential mineral that is recognized for its critical role in athletic performance and overall health [294]. Magnesium helps to improve energy levels, reduce fatigue, and even increase muscle performance, making it a vital nutrient for athletes [36]. With its numerous benefits, magnesium is being increasingly taken by athletes to help them reach peak performance and maintain their physical health [294]. Magnesium helps to improve energy levels by raising the ATP availability, which is best defined as the gold energy stores of cells [283]. Deficiency may cause ATP levels to be depleted, resulting in fatigue and overall reduced performance [34]. The regular consumption of Mg can improve ATP production, providing athletes with increased energy and improved endurance [295]. The mineral is also important for maintaining muscle performance and reducing fatigue [296]. It is known to support muscle contraction and relaxation, allowing for better muscle control and improved performance [142]. It also works to reduce lactic acid buildup in muscles, which may help reduce pain during exercise and improve recovery time [297].

Moreover, Mg has numerous other benefits that support physical wellbeing. It helps to improve sleep quality, regulate blood sugar, reduce stress, and even support the cardiovascular system [294,297]. By regularly taking Mg, athletes may benefit from improved energy production, reduced fatigue, and improved physical health, allowing them to reach their maximum performance potential [36]. The Recommended Dietary Allowance (RDA) is 400 to 420 and 310 to 320 mg/day for 14 to over 70 years of age among males and females, respectively [294].

Magnesium is a versatile mineral that is important for recovery and is found in over 300 enzymes that are involved in energy metabolism [297]. It is linked to strength training and cardiorespiratory processes, showing a reciprocal relationship between exercise and Mg in the human body [142]. Exercise controls Mg distribution and usage [296]. Training triggers Mg to be transferred to areas where energy is produced [37]. For instance, during prolonged activity, serum Mg may be transferred from serum blood to red blood cells (RBCs) or muscle to support exercise. On the other hand, short-term exercise may result in a reduction in the plasma/serum volume and a rise in serum Mg levels [297].

Magnesium contributes to the metabolism of energy and supports typical muscular contraction and relaxation [283]. In male athletes, serum Mg levels are favorably correlated with muscular performance [36]. Additionally, research suggests a possible connection between Mg deficit and muscle cramps by demonstrating how it might alter neuromuscular function [297]. Physically active people might need more magnesium to sustain their peak exercise performance than inactive people do [283]. Low Mg levels may cause ineffective energy metabolism and decreased endurance in individuals who are engaged in a weight training program [294]. Higher Mg consumption is linked to enhanced cardiorespiratory function and lower oxygen demand during aerobic exercise [34]. Most studies reported little impact of 500 mg/day Mg on exercise performance in athletes, unless there is a deficit [37,142,296]. A study of 16 physically active men who were assigned to 300 mg/day for 14 days of Mg supplementation or a control group concluded no direct impact on exercise performance while using Mg supplementation [296].

## 16. Zinc

Athletes of all ages and skill levels rely on zinc (Zn) to keep their bodies performing at their peak [298]. It is an essential mineral that our bodies need for metabolic functions such as cell repair, immune system functioning, hormone production, and healthy skin [38]. Unfortunately, not all athletes receive enough zinc from food intake, which may leave athletes at a disadvantage [38]. According to the UK National Diet and Nutrition Survey (NDNS), current daily intakes are 9.5 mg and 7.6 mg for men and women, respectively [299]. The survey has also shown that 6% of men and 7% of women do not receive enough zinc in their diet, putting them at a greater risk of deficiency [300]. Fortunately, there are many benefits athletes may reap from adding zinc supplements to their routines [300].

One notable benefit of zinc supplementation is the improvement in athletic performance [281], as it reduces blood viscosity and enhances oxygen delivery, thereby boosting aerobic endurance [38,281]. Zinc helps to increase strength and endurance, so athletes may push their bodies to the limit while still receiving the nutrients they need [298]. A double-blinded cross-over study featuring 16 female athletes was conducted to estimate muscle strength and endurance [38]. The participants consumed 135 mg/day of Zn for 14 days and showed remarkably higher dynamic isokinetic strength and angular speed [29]. Additionally, Zn may help to reduce inflammation and soreness, which can accelerate recovery time and reduce the risk of injury [59]. Zinc may also help in improving attention and focus [60]. This can assist athletes in staying focused on their tasks and performing at their best. Zinc is a vital mineral for athletes of all ages and abilities [273]. Including it as a supplement to their regimen may help enhance strength, endurance, and focus while also reducing inflammation and supporting the immune system [92].

In addition, Zn may help to boost the immune system, making it easier for athletes to fight off colds and other illnesses that can stall their progress by increasing neutrophils’ ability to produce ROS after exercise [7]. It has been indicated that Zn oral consumption of 25 mg/day while exercising can reduce exercise-induced changes in immune function to the minimum [298]. Moreover, Zn may help support healthy vision and keep skin healthy, both of which are important for optimal performance [258]. Zn impacts the formation and efficient functioning of the skin and mucous membranes [298]. It helps maintain skin cell membranes, and it plays a part in cell mitosis and differentiation; moreover, it has an essential role in the survival of keratinocytes [215] and even in protecting skin against induced UV radiation damage [301]. Lastly, taking Zn supplements may help athletes meet their nutritional goals without having to increase their caloric intake, making it an ideal supplement for those who are trying to stay lean [6]. Despite Zn supplements being popular among athletes, there is limited proof regarding athletic performance improvement in a period of 1–6 weeks, as shown in the study of Polat, 2011 [302].

To detect the actual impact of zinc oral consumption on the hematological parameters, a study included 24 male kickboxing athletes, who were separated to form the three following groups: the EZ group, meaning they were exercising and consuming 2.5 mg/kg Zn supplement daily; the SZ group, who were supplemented without exercising; and the E group, who were exercising without being supplemented. After the period of 8 weeks, the results showed a significant increase in the erythrocyte count of the EZ group compared to the two other groups (*p* < 0.001). The hemoglobin and hematocrit levels increased in the EZ group (*p* < 0.05). These results revealed that the combination of exercise and Zn supplementation has a beneficial impact on the hematological parameters of athletes, which may result in enhanced performance and increased stamina [303]. Low levels of Zn in the muscles may diminish exercise endurance because it is necessary for the activity of energy metabolism enzymes [304]. Due to the influence of this enzyme on skeletal muscular exercise, lactic dehydrogenase (a Zn-containing enzyme) may facilitate the conversion of lactic acid to pyruvate [38]. This finding contradicts the commonly misinterpreted results of previous studies [59,303], which demonstrated that lactate accumulation does not directly cause fatigue [305].

## 17. Selenium

One possible approach for athletes to achieve their goals is by including selenium (Se) in their diet, as this mineral can be found in a variety of foods [40]. Selenium, when consumed in proper amounts, will help to boost an athlete’s performance, improve mental focus, and reduce inflammation, thereby contributing to overall health and fitness [40,41,257].

Selenium can be found in certain plant and animal products, and it may also be artificially added to processed foods [42]. It is advantageous for athletes due to its powerful antioxidant characteristics that boost the body’s defenses against cell damage, hence increasing endurance, strength, and overall performance [40]. Additionally, Se may increase mental focus, which improves an athlete’s ability to concentrate on tasks and stay motivated, even if the competition gets tougher [41]. It can also help to reduce levels of inflammation and support anti-inflammatory mechanisms, which may boost recovery times and minimize the risk of injuries [95]. Low levels of serum Se are associated with high serum levels of C-reactive protein (CRP), the inflammation biomarker [306] It is well addressed that Se increases glutathione peroxidase production, which prevents the effect of oxidative stress in response to exercise [303]. On the other hand, Se deficiency reduces glutathione peroxidase activity indirectly through controlling the Nuclear Factor kappa-light-chain-enhancer of activated B cells (NF-κB) [257,307]. In the CHIANTI cohort study that assessed coordination performances among 1012 candidates aged 65 years or older, the authors found a reduction in neurological performance that was significantly associated with the low levels of Se [308].

Incorporating Se into an athlete’s diet may be as simple as consuming more foods that are naturally rich in Se or taking it in supplemental form. When consuming the recommended amounts, it may increase the overall health and performance of athletes [306,307,309]. Induced excessive mitochondrial oxidative stress could be caused by Se supplements overdose and may lead to serious health problems [273] marked by organelle damage and dysfunction [310]. Hence, it is important to integrate it into a balanced diet in appropriate doses rather than consuming mega doses [311]. A systematic review of oral Se supplementation of 180 µg/day or 240 µg/day (*selenium methionine*) and 200 µg/day (*Sodium Selenite*) reported a significant drop in lipid hydroperoxide levels and an increase in glutathione peroxidase (GPx) in plasma, erythrocyte, and muscle [306]. The authors concluded that the consumption of Se supplements has no impact on aerobic or anaerobic performance [306]. In addition, the study revealed that Se supplementation may inhibit Se deficiencies induced by high-volume and -intensity exercise, but it has no impact on anaerobic and aerobic athletic performance as well as creatine kinase activity, exercise training-induced adaptations, and testosterone hormone levels [303,306].

## 18. Manganese

As athletes struggle to achieve their best performance, they often look to improve their health and performance. One mineral that has been gaining recognition for its potential benefits is manganese (Mn) [312]. Mn is essential for several bodily functions, including energy production, bone formation, and enzyme activity [264]. Early studies have shown that it may help improve various aspects of athlete health and performance, but little is known about the exact benefits of Mn for athletes [205,303].

Manganese plays an important role in energy production, as it is involved in the breaking-down of carbohydrates, proteins, and fats needed for energy production [205]. It also helps the body to utilize energy more efficiently, which may result in improved endurance during long-term workouts and competitions [304]. Additionally, it aids in the production of important neurotransmitters, which may improve mental focus and coordination during physical activities [304]. Due to its crucial role in bone formation, several studies reported the relationship between Mn and bone health [43] It helps in the development of strong and healthy bones [43], which is crucial for athletes to prevent injury and speed up recovery time. Low serum Mn levels have been reported among osteoporotic women compared to healthy subjects [312]. Studies have also suggested that Mn helps to protect cells from damage caused by ROS, which is important during periods of strenuous exercise [95,304]. It is crucial for scavenging ROS in mitochondrial oxidative stress, as it involves the Mn superoxide dismutase (MnSOD) component [95].

In a clinical trial, it was found that athletes had significantly higher concentrations of basal Mn levels vs. sedentary individuals, as observed through blood and urine measurements [7]. Conversely, sedentary participants exhibited higher urine levels of Mn, which could be attributed to the possibility of iron deficiency in athletes, leading to increased Mn absorption [313]. There is limited evidence regarding Mn and athletic performance; however, athletes should be evaluated periodically for micronutrients deficiencies. Although more research needs to be conducted, the current evidence suggested that Mn may be beneficial for athletes who are looking to optimize their performance and health [205,304]. Adding Mn to an athlete’s diet could be useful for maximizing their performance [95].

## 19. Micronutrients Deficiency and Energy Deficiency’s Impact on an Athlete’s Performance

Pathways for utilizing energy are significantly influenced by vitamins and minerals [93]. A variety of physiological systems depend on micronutrients, which also have an impact on general health and athletic performance [23]. According to the widespread opinion on dietary guidelines for sports, a healthy athlete does not need to exceed RDA values if they consume an adequate number of nutrient-dense foods [267]. Unfortunately, many athletes do not meet the RDA requirements for most micronutrients [6,267]. Micronutrients would logically be impacted by poor macronutrient consumption [25]. It is common among many athletes who are not aware about their exercise energy demands and, on the contrary, suffer from being on a negative energy balance [2]. Negative energy balance due to increased or decreased calorie intake or a combination of both is a powerful disruptor of the endocrine milieu [123]. Additionally, it was associated with increased fatigue and mental disorders [14], reduced fertility, poor bone quality, a higher likelihood of sports injuries, endothelial dysfunction, a poor lipid profile, gastrointestinal disturbances, inflammatory processes, psychiatric conditions (such as emotional state changes/irritability), and poor athletic performance [93,108,154,314].

One common energy deficiency condition among athletes is the female athlete triad, characterized by disordered eating, negative energy balance, and irregular or absent menstrual cycles [315]. This condition predisposes women to menstrual dysfunction (amenorrhea) [316], diminished bone mineral density, and premature osteoporosis [45]. Each defect of the triad represents a significant medical concern, and if occurring together, the possibility of health concerns becomes even more serious and can often cause potential threats to life [317]. Medical adverse consequences associated with disordered eating involve decreased levels of glycogen in its stores, reduced lean body mass, long-term fatigue, dehydration, micronutrient deficiencies, electrolyte and acid-base imbalances, anemia, gastrointestinal diseases, parotid gland enlargement, reduced bone density, and tooth enamel erosion [300]. Osteoporosis can make adolescent female athletes prone to early bone loss and the improper formation of bone, resulting in low bone mineral density [318] and an elevated risk of stress fractures [319]. Bone mineral density lost because of amenorrhea may be totally or, at least partially, irreversible, even with the resumption of menses, calcium supplementation, and estrogen replacement therapy [256,320]. The dispensable role of supplementary vitamins and minerals is a concern of the Dietitians of Canada and the American College of Sports Medicine (ACSM) for ensuring adequate energy requirements are met from a varied and balanced diet with supplementations enrichment. Equally importantly, sports medicine experts may recommend the use of vitamin and mineral supplements in specific conditions such as energy intake restriction, the adoption of a plant-based diet, the presence of illness, or recovery from injuries [242,263,301,321,322]. It is worth noting that individual needs vary, and a personalized approach is crucial when making supplement recommendations.

## 20. Conclusions

Vitamins and minerals are crucial for an athlete’s health and performance, none more so than others. Micronutrients are essential to achieving optimal health and performance. They participate in many metabolic processes in the body, including energy production, muscle growth, and recovery. Athletes need to ensure they consume sufficient quantities of micronutrients to improve their physical activity and performance. A balanced diet that includes a variety of fruits, vegetables, whole grains, and lean proteins may help them meet their micronutrient needs. Additionally, they may benefit from taking a multivitamin supplement if they are not meeting their micronutrient requirements or suffer from malabsorption or specific deficiencies in certain vitamins. However, athletes must avoid taking micronutrient supplements without first ensuring there is a deficiency. It is important to consult with a physician or a dietitian to determine if supplementation is necessary and to obtain a proper prescription.

## Figures and Tables

**Figure 1 sports-11-00109-f001:**
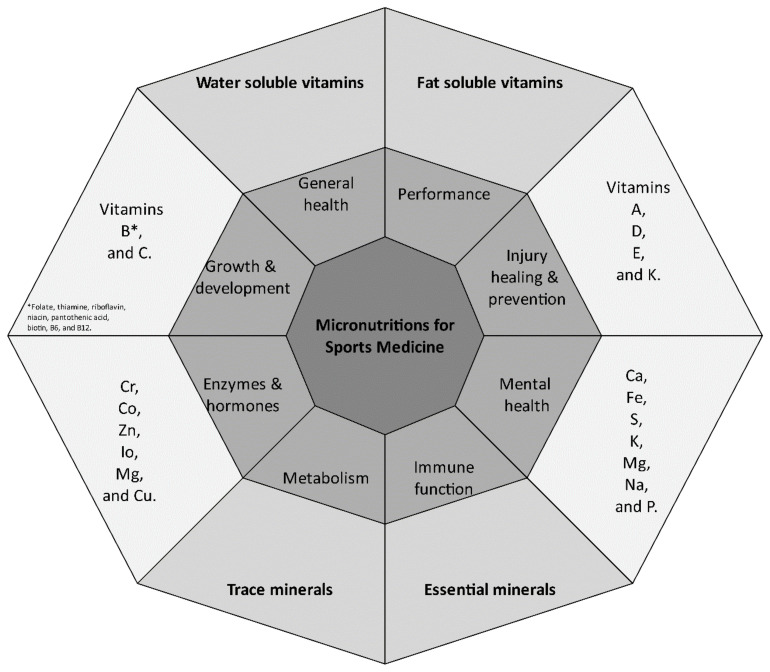
Model of micronutrients’ main functions in sports medicine.

**Table 1 sports-11-00109-t001:** Dietary Difference Intake and Top Sources of Vitamins and Minerals.

Type of Micronutrient	DRI	Top Rich Food Sources	Role in Exercise Performance	Deficiency Effect on Performance
Vitamin A	900 µg for males 700 µg for females [9]	Beef liver Sweet potato Carrot [10]	Protects cells from free radicals’ damage produced during exercise, lowering aches and fatigue. Improves response time and muscle recovery, as it supports protein synthesis, which is a necessity for muscle growth and recovery. Protects against injuries by increasing healing times and promoting the formation of healthy connective tissues [11]	Vitamin A has an oxidation potency which protects athletes against induced and intense exercise free radicals, contributes to the elimination of ROS, and prevents muscle damage and the onset of diseases, despite the higher demand for vitamin A in athletes. Its deficiency is not directly linked to performance impairment, unlike other micronutrients such as iron and others [12].
Vitamin E	15 µg [13]	Sunflower seeds Almonds Cereals ready to eat, RALSTON [14]	Protects body from oxidative stress. Increases the body’s natural immunity and promotes recovery [14] Improves blood flow and increases cardiovascular functions [13]	Decreased performance, recovery, immunity, and blood flow [15]
Vitamin D	1500–2200 IU [16]	Brown mushrooms oily fish, such as salmon, sardines, and cod liver oil [17]	It attenuates inflammation, myopathy, and pain while boosting muscle protein synthesis, ATP percentage, and jump height. Strength, speed, exercise capability, and physical endurance. Musculoskeletal strain avoidance and recovery [18]	Deficient vitamin D concentration seemed to have an unpleasant influence on muscle power, strength, and stamina and elevated musculoskeletal damage, including stress fracture and other injuries impacting inflammation and severe muscle injuries occurring post-intensive exercises [17]
Vitamin K	120 µg for males 90 µg for females [9]	Kale Spinach Parsley [19]	Vitamin k implies its anabolic influence on the bone turnover in several parts including, the regulation of specific gene transcription in osteoblasts, osteoblast differentiation initiation, and activating the bone-associated vitamin k-dependent proteins which play critical roles in the mineralization of the bone matrix and activating vitamin k-dependent proteins, which has an important role in extracellular bone matrix mineralization [19]	Insufficient consumption of vitamin K may be associated with a raised fracture risk [19]
Thiamin (B1)	1.2 mg for males 1.1 mg for females [20]	Fortified breakfast cereals Bacon Sunflower seed [21]	Necessary for the metabolism of amino acids and carbohydrates for nucleic acid precursors development, myelin, and neurotransmitters [21]	Increased oxidative stress [22]
Riboflavin (B2)	1.3 mg for males 1.1 mg for females [23]	Dairy products, meat, egg [21]		Does not have an effect on athletic performance [21]
Niacin (B3)	16 mg for males 14 mg for females [10]	Whole grains, dairy products, milk, and meat [21]	Lowers lipid levels by increasing homocysteine levels [23] Reduces exercise capacity due to blunting fatty acid	May increases exercise capacity [9]
Pantothenic acid (B5)	5 mg [20]		Improves aerobic performance [3]	No proven benefit, so deficiency does not cause any effect [7]
Pyridoxine (B6)	1.3 mg [9]	Fish, beef liver, and other organ meats [24]	Increase muscular growth, strength, and aerobic capacity in the lactic acid and oxygen systems. Relaxing effect increased mental power [25]	No effect
Cyano-cobalamin (B12)	2.4 µg [20]	Fish, meat, poultry, eggs [26]	Essential for the synthesis of DNA and serotonin, required for protein and red blood cell production, enhances muscular mass and blood oxygen carrying capacity, and lessens anxiety [27]	May cause higher odds of anxiety [3]
Folic Acid	400 µg [20]	Dark green leafy vegetables, fruits, nuts, and beans [28]	Crucial for proper brain functioning and works in combination with vitamin b12 in forming red blood cells and supporting iron in performing properly [28]	Megaloblastic anemia, impairing red blood cells, tingling in hands and feet, tiredness, fatigue, weakness, loss of coordination, and weight loss [28]
Vitamin C	90 mg for males 75 mg for females [13]	Citrus fruits, tomatoes, green peppers, kiwifruit [25]	Antioxidant, wound healing and collagen production, boosts energy, and protects from illness and injury [25] Produces collagen, which strengthens joints and muscles [12]	May have a higher chance of getting sick and missing performances; increased wound repair time [13].
Iron (Fe)	8 mg for males [29] 18 mg for females	Lean meat and seafood, nuts, beans [30]	Iron is a key component in a variety of physiological functions that impact athletes’ physical performance. Due to the number of iron-dependent proteins and enzymes affecting energy production in mitochondria and the oxygen delivered to the muscles. Oxygen-carrying capacity and mitochondrial oxidative phosphorylation activity, which is determined by the hemoglobin mass, skeletal muscle, and maximal oxygen consumption [15]	Iron deficiency, whether combined with anemia or not, can lead to muscle impaired function and limited endurance capacity, which affect athletic performance and training adaptation negatively [31].
Calcium (Ca)	1500 mg [32,33]	Dairy products, sardines and salmon, kale, broccoli [34]	Since calcium ion could move through and out of a cell’s cytoplasm, it has an essential role in signaling for various process in the cell, including muscle contraction, exocytosis, proliferation of action potentials through cardiac muscle, and neurotransmitter release [35]	Calcium plays a crucial role in maintenance, growth regulating muscle contraction, normal blood clotting, and the conduction of nerve and bone tissue repair. The possibility of stress fracture and low bone-mineral density is elevated by low available levels of energy. In certain cases, such as female athletes, insufficient calcium intake combined with menstrual dysfunction increases the risk ratio [36]
Potassium (K)	3500 mg for males [36] 2500 mg for females	Dried fruits (raisins, apricots) Beans, lentils Potatoes [37]	Potassium is a major source of energy for athletes. It helps to reduce the amount of lactic acid stored in the muscles, which can lead to fatigue as well as maintain a healthy metabolism. Potassium is also involved in the breakdown of carbohydrates, which helps to keep energy levels high during intense physical activity [37]	Whole body and muscle fatigue such as inappropriate exercise performance could be the result of the acute depletion of the trans-sarcolemma k+ [36]
Magnesium (Mg)	400 mg for males 310 mg for females [38]	Whole grains and dark-green, leafy vegetables, dried beans, and legumes [39].	Magnesium (Mg) is an important mineral with an essential impact on the human body. It plays a crucial part in maintaining proper muscle functioning and energy metabolism; several studies assessed the relation between Mg status/supplementation and exercise performance and found a direct correlation between magnesium demand and raised levels of physical activity [38]	Athletes who are insufficient in magnesium levels are not protected from inflammatory reactions, which may increase the risk of hypertension, atherosclerosis, diabetes mellitus, osteoporosis, and cancer occurrence [39]
Zinc (Zn)	8 mg for males 11 mg for females [32]	Meat, fish, seafood [40]	Zinc has an advantageous effect on performance improvement by participating in muscle energy production, recruiting fast twitch muscle fibers and protein synthesis, which is necessary for physical performance [40]	Deficient zinc levels in athletes reduced endurance, led to a significant reduction in body weight, and latened fatigue with impaired endurance and osteoporosis risk [41]
Selenium (Se)	55 mg [42]	Brazil nuts, seafoods, and organ meats [43]	Selenium in glutathione peroxidase aids in allergies and inflammatory diseases elimination, defending the muscles and the cardiovascular system [43]	Insufficient Se levels may raise exercise-induced oxidative stress over time [44]
Manganese (Mn)	2.3 mg for males 1.8 mg for females [45]	Whole grains, oysters, mussels, nuts [46]	Mn is an essential nutrient necessary for energy metabolism and in antioxidant enzymes that protect cells from damage due to free radicals [46]	The deficiency of manganese was indicated as an etiological agent in joint diseases and hip abnormalities development [45]

Abbreviations: µg: microgram; DNA: deoxyribonucleic acid; IU: international unit; mg: milligram; ROS: reactive oxygen species.

**Table 2 sports-11-00109-t002:** Comprehensive overview of articles about micronutrients and sport performance.

SN	Author	Year	Study Design	Micronutrient	Source of the Micronutrient	Sample Size	Mean Age	Gender	Conclusion
1	Bezuglov E. [47]	2023	Cohort	Vit D	Natural source	68	M = 18.2 ± 1.9 F = 17.3 ± 2.6	M = 23 F = 45	No correlation was found between serum Vit D level and strength, speed characteristics, total testosterone concentration, performance in the 20 m and 30 m sprint, countermovement jump, and broad jump.
2	Pallante P.I. [48]	2023	Clinical trial	Vit D, Mg, and Zn	Natural source	30	F = 18–22	F = 30	No association was found between Mg and Zn intake and PMS. However, lower Vit D intake tended to be associated with presenting PMS in female athletes.
3	Zhang J. [49]	2023	Cross-sectional	B Vits (B1, B2, B3, B5, B6, B9, and B12)	Supplement	427	All = 27.65 ± 3.78 M = 27.47 ± 3.87 F = 27.81 ± 3.70	M = 210 F = 217	The supplementation of B vitamins and pectin may be beneficial for exercise performance and post-exercise recovery.
4	Mastali V.P. [50]	2022	Quasi-experimental	Vit D	Supplement	24	Group 1 = 24.33 ± 2.7 Group 2 = 25.83 ± 3.18	M = 24	Short-term Vitamin D supplementation could prevent myocytes and hepatocytes damage induced by EAE.
5	Martínez-Ferrán M. [51]	2022	Double-blind randomized controlled trial	Vit C and Vit E	Supplement	18	Group 1 = 47.90 ± 5.75 Group 2 = 46.76 ± 4.60	M = 18	Vit C and E supplementation did not seem to help with EIMD in endurance-trained individuals.
6	AL-Qurashi T.M. [9]	2022	Experimental trial	Mineral water	Natural source	20	M = 21.7 ± 3.21	M = 20	Rehydration with mineral water such as zamzam is unlikely to impair cardiorespiratory fitness, even with an intake equal to 100% of the loss in body weight.
7	Mesquita E.D.D.L. [10]	2022	Cross-sectional	Vit D	Natural source	75	N/A	N/A	Only adolescents with a combination of sports participation and higher serum concentrations of Vit D showed better bone geometry, indicating the relevance of the combination of both factors to bone health.
8	Chen L.-Y. [11]	2022	Cross-sectional	Vit D	Supplement	50	M = 21.14 ± 1.95	M = 50	Vit D may play a significant role in cardiovascular function that influences endothelial and smooth muscle cell function. Vit D deficiency may increase the risk of incident cardiovascular events after acute exhaustive exercise, even in healthy and active adults.
9	Książek A. [13]	2022	Cross-sectional	Vit D	Natural source	40	Group 1 = 22.5 ± 2.9 Group 2 = 21.9 ± 3.7	M = 40	There was a significant correlation between Vit D metabolites and handgrip strength and vertical jump variables in indoor players.
10	Sone R. [14]	2022	Experimental trial	Antioxidant	Supplement	7	M = 22.6 ± 1.3	M = 7	Mineral-rich antioxidant supplements did not directly affect oxidative stress markers in the blood but suggested that performance (lactate) and salivary nitric oxide could be improved.
11	Marley A. [16]	2022	Randomized doubled-blind controlled trial	Vit D3	Supplement	27	M = 25 ± 5	M = 27	Supplementing 50,000 IU of Vit D3 per week for six weeks combined with six weeks of SIT may improve markers of aerobic and anaerobic performance in recreational male combat sport athletes.
12	Rockwell M.S. [17]	2022	Cross-sectional	Vit D	Natural source	53	M = 19.9 ± 0.4 F = 19.7 ± 0.3	M = 25 F = 28	The bioavailable Vit D concentration was associated with a higher total, ap spine, and hip bone mineral density (bmd), but the total Vit D concentration was not related to total bmd and was negatively associated with ap spine and hip bmd
13	Brzeziański M [17]	2022	Experimental trial	Vit D3	Supplement	25	M = 17.5 ± 0.7	M = 25	Vit D3 supplementation in a dose of 6000 IU/d increased its serum concentration in all the study groups of young athletes, causing the equalization of a suboptimal supply of Vit D3 in the serum. The study was not able to prove the ability of Vit D3 supplemented in the proposed dose to influence IL-6 or CRP concentrations in athletes.
14	de Brito E. [52]	2022	Randomized clinical trial	Vit C and Vit E	Supplement	14	All = 26.2 ± 5	N/A	The association of Vits (C and E) with cryotherapy attenuated the inflammatory response and pain, favoring recovery after an acute resistance exercise session.
15	Şenışık S. [26]	2022	Cross-sectional	Vit D	Natural source	256	All = 13.2 ± 2.2	N/A	The frequency of Vit D deficiency and insufficiency is higher in indoor athletes and is especially associated with the risk of bone injuries
16	Most A. [30]	2021	Cross-sectional	Vit D	Natural source	112	M = 26.1 ± 5.2	M = 112	Vit D insufficiency was associated with lower maximal aerobic power, as assessed with a standardized exhaustive cycling ergometer test. The Vit D level was the only independent predictor of maximal aerobic power in these athletes, highlighting the impact of Vit D on physical performance. A Vit D level of less than 30 ng/mL should be maintained to ensure optimal physical performance in these athletes.
17	Nikniaz L. [47]	2021	Randomized controlled trial	Vit D	Supplement	40	M = 30.40 ± 4.08	M = 40	Aerobic exercise combined with Vit D supplementation can reduce serum inflammatory factors and anti-inflammatory proteins and improve lung function after four weeks of intervention. The combination of aerobic exercise and Vit D supplementation remarkably reduced TNF-α, IL-6, and CC16. Aerobic exercise alone and the combination of aerobic exercise and Vit D supplementation significantly increased FEV1 and FVC.
18	Kawashima I. [53]	2021	Prospective cohort study	Vit D	Supplement	42	M = 20 ± 1	M = 42	Vit D supplementation of 25 μg/day significantly increased the serum Vit D level in elite male collegiate athletes. Vit D supplementation may play a role in maintaining athletes’ body fat percentage under circumstances where sports activity has decreased.
19	Mieszkowski J. [50]	2021	Cross-sectional	Vit D	Natural source	32	Group 1 = 20.6 ± 3.3 Group 2= 19.9 ± 1.0	M = 32	Vit D metabolites affect the anaerobic performance and bone turnover markers at rest and after exercise
20	Janssen J.J.E. [54]	2021	Experimental trial	Vit B2	Natural source	31	Group 1 = 24.0 Group 2 = 21.8	F = 31	A single bout of exercise significantly increased egr activity but did not affect egrac values, indicating that a single bout of exercise did not affect Vit B2 status.
21	Ali A. [55]	2021	Clinical trial (crossover study)	Vit C	Natural source (sungold kiwifruit)	10	F = 30.92 ± 7.32	F = 10	Consuming liquid Vit C prior to high-intensity cycling appears to be more effective than eating kiwifruit in ameliorating exercise-induced stress in recreationally active women of reproductive age
22	Valtueña J. [56]	2021	Longitudinal study	Vit D	Natural source and supplement	95	M = 27.3 ± 4.6	M = 95	A positive interaction with supplementation existed in two different directions; outdoor training improves the Vit D status only in supplemented team players, and supplementation has a positive influence on the Vit D status only in individuals with adequate sun exposure. Vit D deficiency might affect team players’ overall health and performance
23	Wilson-Barnes S.L. [57]	2021	Longitudinal study	Vit D	Natural source	50	Group 1 = 20.8 ± 1.9 Group 2 = 24.8 ± 4.2	M = 24 F = 26	Predictors of physical performance were not associated with Vit D status within both groups or during both seasons.
24	Kamińska J. [58]	2021	Randomized controlled trial	(Ca^2+^, Na^+^, Mg^2+^, K^+^, Hco^3−^, So4^2−^, Cl^−^, and F^−^)	Mineral in the fluids	14	F = 21.9 ± 2.3	F = 14	The osmolarity of consumed fluids does not significantly affect the indicators of the water–electrolyte balance and the acid–base balance during exercises; such an effect is only noticeable after consuming an isotonic drink. The degree of mineralization of the water consumed by female field hockey players did not affect the indicators of the water–electrolyte and acid–base balance in the blood and urine.
25	Kawashima I. [53]	2021	Cross-sectional	Vit D	Natural source	48	M = 19.8 ± 0.9	M = 48	Vit D is insufficiently widespread among indoor elite athletes, with the majority of them suffering from Vit D deficiency. Outdoor players had a sufficient Vit D level. Vit D insufficiently had significantly higher body fat percentages than sufficient Vit D athletes.
26	Alves J. [59]	2021	Cross-sectional	Antioxidant Vits (Vit C and Vit E)		84	M = 23.2 ± 3.25	M = 84	A maximum incremental test did not produce any changes in plasma Vits in athletes. However, it increased the levels of Vit C in erythrocytes and decreased malondialdehyde values in plasma and Vit E in erythrocytes. The levels of malondialdehyde, Vit C, and Vit E were related to performance parameters.
27	Martusevich A.K. [60]	2021	Randomized controlled trial	Vit-Mineral Complex (Vits C, E, A, D, Group B, Minerals and Trace Elements, Β-Carotene (1.5 Mg), Lutein (4.5 Mg), and L-Carnitine)	Supplement	74	N/A	N/A	The use of an individually prescribed Vit-mineral complex may allow for optimizing the state of the oxidative metabolism of athletes’ blood plasma.
28	Sasaki C.A.L. [61]	2021	Cross sectional	All Micronutrient	Natural Source and Supplement	101	All = 33.32 ± 9.88	M = 82 F = 19	The prevalence of inadequacy for Vit D, calcium, Vit A, thiamine, riboflavin, and zinc was significantly higher in para-athletes. The prevalence of the risk for iron deficiency was recorded in female para-athletes.
29	Marley A. [16]	2021	Single-blind crossover	Vit D3	Supplement	27	M = 24 ± 4	M = 27	Given the endurance adaptations from Vit D supplementation and the importance of endurance for combat performance, recreational combat athletes should supplement at 50,000 IU per week for six weeks. There is no additional benefit of increasing the dose above 50,000 IU Vit D per week.
30	Pilch W. [62]	2020	Randomized control trial	Vit D	Supplement	60	M = 20–24	M = 60	The plasma Vit D level is considered a significant indicator for reducing muscle cell damage induced by eccentric exercise.
31	Crewther B. [63]	2020	Cross-sectional	Vit D	Natural source	88	N/A	M = 88	Serum Vit D was a poor predictor of exercise performance, but it did moderate (with cortisol) the testosterone link to muscle power.
32	Shalaby M.N. [64]	2020	Experimental trial	Vit D3	Supplement	20	All = 18.64 ± 0.43	N/A	Vit D supplementation may affect the muscle function and health of the athlete. It is recommended that Vit D levels should be checked on an annual basis in all athletes for their health.
33	Aminaei M. [65]	2020	Quasi-experimental	Vit D and Calcium	Supplement	40	F = 28.1 ± 2.7	F = 40	Eight weeks of TRX training with Vit D and calcium supplementation improved BMI and HDL serum levels. The intensity and duration of training and supplementation probably have positive effects on lipids profiles.
34	Bauer P. [66]	2020	Cross-sectional	Vit D	Supplement	120	M = 26 ± 5	M = 120	Athletes with sufficient Vit D achieved a higher maximum systolic BP and a higher maximum power output. Better performance was recoded among athletes with sufficient Vit D.
35	Molina-López J. [67]	2020	Clinical trial	MultiVit\Mineral; Vit A, Vit E, Vit C, Vit B, Vit B2, Vit B6, Vit B12, Vit D, Biotin, Floate, Niacin, Pantothenic Acid, Calcium, Phosphorous, Magnesium, Iron, Iodine, Cooper, Manganese, Selenium, Zinc	Supplement	26	Group 1 = 22.9 ± 2.7 Group 2 = 20.9 ± 2.8	M = 26	Elite handball athletes showed a different expression profile in reference to key genes implicated in several sports’ performance-related functions compared to the sedentary controls, in addition to the modulation of gene expression after multiVit/mineral supplementation.
36	Krzywański J. [68]	2020	Clinical trial	Vit B12	Injection	243	Group 1 = 23 ± 1 Group 2 = 25 ± 1	N/A	A weak positive correlation between the Vit B12 concentration and Hb and between MCH and Hct were found. Higher values of hemoglobin and hematocrit were observed after B12 injections in endurance athletes.
37	Sekel N.M. [69]	2020	A quasi-experimental trial	Vit D	Supplement	20	All = 20.25 ± 0.85	M = 7 F = 13	A daily intake of 10,000 IU Vit D was not sufficient to recover the deficiency, while it protected against seasonal declines. An intake of 5000 IU daily was insufficient and failed to attenuate against seasonal decline.
38	Millward D. [70]	2020	Cohort study	Vit D	Supplement	802	M = 18.7 ± 1.2 F = 18.6 ± 1.2	M = 498 F = 305	Correcting low serum Vit D levels reduced the risk of stress fracture.
39	Pradita D.K. [71]	2020	Cross-sectional	Iron	Natural source	70	F = 12–21	F = 70	Significant relationships were observed between iron deficiency based on serum ferritin and muscle mass with bone density in young female athletes.
40	Shafiee S.E. [72]	2020	Cross-sectional	Vit D	Supplement	100	M = 28 ± 6	M = 100	Lower serum concentrations of Vit D are associated with the risk of ACLI in male athletes.
41	Ashtary-Larky D. [73]	2020	Non-randomized crossover design	Vit D	Injection	14	M = 24.3 ± 2.4	M = 14	Among resistance-trained males who suffer from Vit D deficiency, a single injection of Vit D reduced the insulin concentration and blood glucose levels and improved insulin resistance. Single injections of 300,000 IU Vit D had no impact on muscle damage improvement and inflammatory response. Improvement in the Vit D level had no impact on the resting metabolic rate nor inflammatory and cardiovascular biomarkers.
42	Farapti F [74]	2019	Cross-sectional study	Vit E, Vit C, Vit A, Zn, and Cu	Natural source	40	All = 23.08 ± 4.32 M = 22.96 ± 3.98 F = 23.29 ± 5.03	M = 26 F = 14	A low intake of nutrients might deplete the Vit/mineral status, especially in Vit C status among female athletes.
43	Kim D.K. [75]	2019	Retrospective study	Vit D	Natural source	52	M = 23.2 ± 4.5	M = 52	Although Vit D insufficiency was not associated with isokinetic muscle weakness, monitoring its levels is very important for musculoskeletal health; especially, the deficiency was common among elite volleyball players.
44	Bauer P. [76]	2019	Cross-sectional study	Vit D	Natural source	50	M = 26 ± 5	M = 50	Vit D insufficiency was associated with a significant increase in central systolic and diastolic blood pressure among elite athletes.
45	Schaad K.A. [77]	2019	Retrospective review of records	Vit D	Natural source	381, 818	All = 18–64	M = 329, 085 F = 52, 733	Individuals living in a northerly latitude might be more prone for Vit D deficiency and at a higher risk for the diagnosis of depression. Vit D status monitoring was very important among military athletes’ members to prevent the risk for depression.
46	Seo M.-W. [78]	2019	Cross-sectional study	Vit D	Natural Source and Supplement	47	M = 16.7 ± 0.84	M = 47	Anaerobic capacity was correlated poorly with Vit D status. Mechanisms were not clear for how Vit D influence anaerobic performance. During growth periods, it is important to consider the importance of Vit D regarding health benefits.
47	de Oliveira D.C.X. [28]	2019	Double-blind, controlled clinical trial	Vit C and Vit E	Supplement	21	M = 19.9 ± 0.3	M = 21	Antioxidant supplementation reduced oxidative stress only among young athlete football players. It has no ergogenic aids on muscle damage or muscle soreness due to acute exercise.
48	Higgins M.F. [79]	2019	A double-blind, placebo-controlled crossover trial	Mineral	Deep ocean mineral	9	M = 22 ± 1	M = 9	The deep ocean mineral content of minerals and trace elements had many health recovery benefits for active male soccer players who have a prolonged high-intensity running capacity in thermoneutral environmental conditions.
49	Aydın C.G. [80]	2019	Cross-sectional study	Vit D	Natural source and supplement	555	All = 5–52	M = 229 F = 326	Participating athletes’ performance benefited from improving Vit D levels, especially during the winter season.
50	Alkoot M.J. [81]	2019	Cross-sectional study	Vit D	Supplement	250	M ≥ 21	M = 250	Vit D deficiency was common among professional athletes. Eight winter weeks of supplementation with cholecalciferol increases Vit D serum levels, but not enough for professional athletes.
51	Michishita R. [82]	2019	Cross-sectional	Sodium, Potassium, and Vit E	Natural source	302	All = 48.4 ± 11.3	M = 64 F = 238	Subjects who do not suffer from hypertension diseases would benefit from the dietary sodium, potassium, and antioxidant Vit intake.
52	Bauer P. [83]	2019	Prospective, non-interventional study	Vit D	Supplement	70	N/A	M = 70	Professional handball athletes suffered from Vit D insufficiency even in summer. Their insufficiency level negatively impacted their physical performance, which is a risk for musculoskeletal injuries and infections.
53	Umarov J. [84]	2019	Prospective, non-interventional, observational	Vit D	N/A	70	N/A	N/A	Vit D insufficiency is common among elite athletes engaged in synchronized swimming and swimmers. It is accompanied by a decrease in ifn-γ, an increase in tnf-α, IL-4, and IL-6 levels, and an elevation of urti morbidity. Seasonal monitoring and correction of the Vit D level for the normalization of the cytokine profile and a decrease in urti morbidity is definitely advised
54	Wrzosek M. [85]	2019	Cross-sectional	Vit D, Calcium	Natural source	593	F = 18–50	F = 593	It is important to educate women about the necessity to provide the body with proper calcium and Vit D intake levels in a diet to avoid health problems, resulting from the deficit of the nutrients.
55	Alimoradi K. [27]	2019	Randomized controlled clinical trial	Vit D	Supplement	70	Group 1 = 24.09 ± 5.06 Group 2 = 22.71 ± 4.07	M = 36 F = 33	Weekly supplementation with 50,000 IU Vit D resulted in a nearly 17 ng/mL increment in circulating calcidiol. This increase was associated with a significant improvement in power leg press and sprint tests in D-supplemented group.
56	Carswell A.T. [86]	2018	Study 1: prospective cohort Study 2: double-blind, randomized, placebo-controlled trail	Vit D	Natural sources	Study 1 = 967	Study 1 = 22 ± 3 Study 2 = 22 ± 3	Study 1: M = 621; F = 364 Study 2: M = 173	The Vit D status was associated with endurance performance but not strength or power in a prospective cohort study. Achieving Vit D sufficiency via safe, simulated summer sunlight or oral Vit D3 supplementation did not improve exercise performance.
57	Jung H.C. [87]	2018	Double-blind, randomized, and placebo-controlled design	Vit D3	Supplement	Study 2 = 173	All = 20.1 ± 0.15	N/A	Correcting Vit D insufficiency improves some but not all aspects of performance. Thus, the efficacy of Vit D supplementation to enhance performance remains unclear.
58	Orysiak J. [88]	2018	Cross-sectional	Iron, Vit D	Natural source	35	M = 17.2 ± 0.9	M = 50	Vit D insufficiency is highly prevalent in ice hockey players, but the Vit D level was not associated with exercise performance or indices of iron status.
59	Malczewska-Lenczowska J. [89]	2018	Cross-sectional	Iron, Vit D	Natural source	50	F = 20.0 ± 4.4	F = 219	The association between Vit D and iron status in female athletes is complex, and it is challenging to determine which nutrient exerts a stronger influence over the other.
60	Radzhabkadiev R.M. [90]	2018	Cross-sectional	Vit A, B, B1, B2, C	Supplement	18	M = 21.7 ± 0.8 F = 23.1 ± 1.5	M = 92 F = 67	To maintain the optimal Vit status of the athlete’s organism, it was inappropriate to use excessive doses of Vits C (>200–300 mg/day), E (>50 mg TE/day), and A (>1500 μg RE/day).
61	Wei C.-Y. [91]	2017	Double-blind placebo-controlled crossover	Trace element, Deep Ocean Mineral (Dom)	Supplement (dom)	159	Group 1 = 21.2 ± 0.4 Group 2 = 46.8 ± 1.4	M = 21	Minerals and trace elements from deep oceans possess great promise in developing supplements to increase the cerebral hemodynamic response against a physical challenge and during post-exercise recovery for middle-aged men.
62	DiSilvestro R.A. [92]	2017	Randomized controlled trial	Minerals (Iron, Zinc, Copper)	Supplement	21	F = 18–30	F = 76	A combination of micronutrients can improve aerobic exercise performance in one set of circumstances.
63	Wardenaar F. [93]	2017	Cross-sectional	All Vits and minerals	Supplement and Natural Source	26	M = 23.5 ± 11.5 F = 22.0 ± 7.6	M = 327 F = 226	Both users and non-users of nutritional supplements reported inadequate intake of micronutrients. For most micronutrients, the use of nutritional supplements does not completely compensate for intakes below recommendation.
64	Backx E. [94]	2017	A longitudinal study	Vit D	Supplement	553	All = 22 ± 4	M = 22 F = 30	A sufficient Vit D concentration in summer did not guarantee a sufficient status in winter. Coaches and medical professionals should monitor athletes’ Vit D concentration regularly to prevent Vit D deficiency.
65	Owens D.J. [94]	2017	Randomized clinical trial	Vit D	Supplement	52	M = 26 ± 3	M = 46	Frequent low doses of Vit D intake and gradual supplementation withdrawing were more preferable than the opposite.
66	Dahlquist D.T. [95]	2017	Randomized, placebo-controlled, single-blinded, triple-crossover study	Vits D3 and K2	Supplement	46	M = 26.9 ± 6.4	M = 10	Vit D and K2 had no significant impact on hepcidin-25, IL-6, Hb, hematocrit, serum ferritin, or serum iron.
67	Nayir T. [96]	2017	Cross-sectional study	Vit D	Natural source	10	N/A	M = 679 F = 447	Vit D insufficiency is common in long-lasting sports.
68	Cheng-Shiun He,. [97]	2016	Randomized controlled trial	Vit K2	Supplement	76	N/A	N/A	Vit K2 supplementation had been reported to improve cardiovascular function in diseased patients.
69	Hildebrand R.A. [98]	2016	Cross-sectional study	Vit D	Serum 25-oh d	1126	All ≥ 18	N/A	Vit D insufficiency and deficiency were common among collegiate athletes, which affect their muscular strength and power. It is important to consider the benefits of Vit D for optimal training to maximize performance, especially in muscular strength anaerobic power exercise.
70	Cassity E.P. [99]	2016	Randomized controlled trial	Vit D	Supplement	113	All ≥ 18	M = 19 F = 13	There was an inverse correlation between BMI and 4000 IU of Vit D supplementation in athletes. Normal-BMI athletes demanded less than the upper limit of Vit D supplementation to sustain sufficient Vit D status. High-bone-turnover athletes lost a significant amount of Vit D during training.
71	Darr R.L. [100]	2016	Double blinded, randomized clinical trial	Vit D3	Supplement	32	Group 1 = 42.0 ± 10.7 Group 2 = 36.4 ± 6.9	M = 13	Vit D supplementation post-low or -moderate exercise but not resistance exercise enhances IGFBP3, which promotes the delivery of IGF1 to tissues.
72	Krzywanski J. [101]	2016	Retrospective	Vite D	Sun Exposure and Supplement	13	N/A	M = 228 F = 181	Polish elite athletes suffer from an insufficient Vit D status that affecst their health and performance negatively, especially among indoor sports. Hence, it is recommended to be exposed to sunlight combined with an oral Vit D supplement.
73	Capó X. [102]	2016	Controlled clinical trial	Vit E	Beverage supplementation	409	Group 1 = 22.8 ± 3.8 Group 2 = 45.6 ± 1.6	M = 10	To improve the pro-inflammatory circulating in young athletes, functional beverage supplementation is recommended during exercise.
74	Backx E.M.P. [103]	2016	Randomized, double blind, dose-response study	Vit D	Supplement	10	All = 8–32	M = 54 F = 48	Intake of 2200 IU/day of Vit D can recover the deficiency among athletes.
75	Todd J. [104]	2016	Cross-sectional	Vit D	Supplement	128	All = 25 ± 5	M = 46 F = 46	Irish athletes recovered from Vit D insufficiency due to supplementation program-targeted elite sports.
76	Maruyama-Nagao A. [105]	2016	Cross-sectional	Vit D	Sunlight exposure	92	F = 20–22	F = 30	Vit D insufficiency is common among indoor sports athletes compared to outdoor sports athletes, mostly during winter, which could influence bone mineralization over the three months.
77	Wyon M.A. [106]	2016	Randomized, placebo-controlled, double-blind trial	Vits D3 and K2	Supplement	30	Group 1 = 29.6 ± 10.6 Group 2 = 26.6 ± 7.4	M = 22	Among elite indoor athletes who suffer from insufficient Vit D, a shot of 150,000 IU showed a favorable impact on muscle function and Vit D level.
78	Keen D.A. [107]	2016	Randomized experimental study	Multimineral	Deep ocean mineral water (dom)	22	All = 23 ± 1.2	N/A	The intake of kona showed a positive impact on exercise performance, especially when consumed during the recovery strategy of rehydration and post-exercise.
79	Veskoukis A.S. [108]	2016	A randomized, placebo-controlled, double-blind trial	Vit C	Supplement	8	M = 21.1 ± 3.1	M = 30	The intake of Vit C at rest showed a positive effect on reducing the oxidative stress, enhancing the antioxidant capacity, and altering the redox state and inflammation biomarkers.
80	Heffler E. [109]	2016	Cross-sectional	Vit D	Supplement and Natural source	30	All = 13–25	M = 24 F = 13	Vit D deficiency negatively affected athletic performance by altering calcium homeostasis among young athletes who lived above the 40th parallel north.
81	Kaul A. [110]	2016	Cohort	Vit D	Natural source	37	All = 20–79	M = 677 F = 700	Vit D levels correlated positively with physical performance among healthy subjects.
82	He C.-S. [111]	2016	N/A	Vit D	N/A	1377	N/A	N/A	Sunlight exposure in the summer combined with everyday supplementation of 1000 iu in the winter is the practical proposal to achieve Vit D sufficiency.
83	He C.-S. [111]	2016	Randomized controlled trial	Vit D3	Supplement	N/A	M = 20.4 ± 1.9	M = 39	To improve respiratory infections resistance, it is recommended that athletes consume a daily dose of 5000 IU Vit D supplement by monitoring the expression of sIgA and Cathelicidins.
84	Chenoweth L.M. [112]	2015	Randomized, single-blinded, crossover design study	Vit C, Vit E, Zinc, Selenium, Copper, and Manganese	Supplement	39	All = 22 ± 1	M = 5 F = 5	Among subjects who do not eat enough servings of fruits and vegetables, the intake of Vits and minerals supplements during exercise increased TAS, improved resting expiratory flow rates, and reduced EFL during exercise.
85	Caruana H. [113]	2015	Cross-over study	Vit C	Supplement	10	M = 21.1 ± 0.84	M = 10	In young men, hyperemia post-strenuous muscle contraction is reduced by ROS.
86	Fitzgerald J.S. [114]	2015	A cross-sectional design	Vit D	Serum level	10	M = 20.1 ± 1.5	M = 53	Among young male ice hockey players, the Vit D level was positively associated with the strength of the upper body but not the lower body force or the production of power.
87	Veasey R.C. [115]	2015	Placebo-controlled, double-blind, randomized, balanced cross-over study	Vit D	Supplement	53	M = 21.4 ± 3.0	M = 40	Pre-moderate intensity exercise and the intake of Vit and mineral complex supplements with guarana promote memory performance in active males.
88	Price O.J. [116]	2015	Single-blind, placebo-controlled trial	Vit D	Supplement	40	All = 35 ± 8	M = 9 F = 1	Vit D and omega-3 PUFA supplementation does not ease the reduction in lung function post-EVH.
89	Allison R.J. [117]	2015	Cross-sectional	Vit D	Supplement and Natural source	10	Group 1 = 23.9 ± 5.0 Group 2 = 23.9 ± 4.4 Group 3 = 24.6 ± 4.6 Group 4 = 25.2 ± 4.6	M = 950	Among male athletes, no association was found between Vit D and BMD and T-score.
90	Allison R.J. [118]	2015	Cross-sectional	Vit D	Natural source	950	Group 1 = 21.7 ± 4.8 Group 2 = 22.3 ± 5.2 Group 3 = 24.0 ± 5.5 Group 4 = 23.3 ± 5.3	M = 750	Severe Vit D-deficient athletes show significantly fewer cardiac structural parameters than insufficient and sufficient athletes.
91	Heller J.E. [119]	2015	Cross-sectional	Vit D	Supplement and natural source	750	All = 20.7 ± 1.6	M = 24 F = 18	Athletes with a large body size and/or excess adiposity may be at a higher risk for Vit D insufficiency and deficiency.
92	Popovic L.M. [120]	2015	Clinical trial	Vit C	Supplement	42	Group 1 = 22.5 ± 1.5 Group 2 = 24.5 ± 2.5	M = 60	Vit C supplementation can suppress the lipid peroxidation process during exercise but cannot affect the neutrophil inflammatory response in either exercise group.
93	Turchaninov D.V. [121]	2015	Clinical trial	N/A	Fortified fermented milk product	60	All = 12–17	N/A	It is recommended among sport-active adolescents to improve their intake with the dairy products “bifidin” and “prolacta” hypoVitosis and micro-elementoses.
94	Díaz V. [122]	2015	Clinical trial	Vit C, Vit E	Supplement	94	M = 26.9 ± 6.7	M = 10	Intakes of 500 mg/day Vit C and 400 IU/day Vit E for 28 days had no effect on the hepcidin level due to inflammatory and iron signals.
95	Theodorou A.A. [123]	2014	Clinical trial	Vit C	Supplement	10	Group 1 = 2.6 ± 0.9 Group 2 = 22.8 ± 1.1	M = 20	Oxidative stress is decreased post-exercise, while it is increased with antioxidant stimulus exposure.
96	Fitzgerald J.S. [124]	2014	Cross-sectional	Vit D	Natural source	20	M = 20.1 ± 1.5	M = 53	Throughout the skate treadmill gxt, the level of Vit D was not significantly assoiciated with any physiological or physical parameter.
97	Karakilcik A.Z. [125]	2014	Cross-sectional	Vit C	Supplement	52	M = 23.50 ± 0.59	M = 22	In young soccer players, the intake of Vit C combined with exercise decreases TBARS-levels and may affect the values of PLT, MPV, PCT, and RDW.
98	Koundourakis N.E. [126]	2014	Cross-sectional	Vit D	N/A	22	M = 25.6 ± 6.2	M = 67	The concentration of Vit D was significantly associated with the pre- and post-experimental performance parameters.
99	Aguiló A. [127]	2014	Double-blinded study	Vit C	Supplement	67	Group 1 = 39.5 ± 5.6 Group 2 = 37.2 ± 5.4	M = 31	In non-exhaustive exercise, the intake of 250 mg Vit C twice a day for 15 days had no impact on IL-6 and IL-10, while it increased the production of IL-6 and IL-10 during the 2 h post-exercise recovery.
100	Shanely R.A. [128]	2014	Cross-sectional	Vit D	Supplementation with Portobello Mushroom Powder	31	Group 1 = 15.9 ± 0.29 Group 2 = 16.6 ± 0.23	M = 33	Among high school athletes, the intake of 600 IU/day of Vit D raised the Vit D level without any impact on muscular function or damage post-exercise.
101	Beketova N.A. [129]	2014	Cross-sectional	Vits A, E, C, B2, and Beta-Carotene	Supplement	33	Group 1 = 18.5 ± 0.3 Group 2 = 26.8 ± 0.7	N/A	It is important to enrich all athletes’ diets with supplements, considering age and gender variation.
102	Soria M. [130]	2014	Prospective, simple blind, placebo-controlled trial	Sulfur	Sulfurous mineral water (smw)	169	M = 27.3 ± 4.1	M = 30	To prevent muscle damage post-exercise, it is recommended to intake smw supplements for 3 weeks.
103	Barker T. [131]	2014	Cross-sectional	Vit D	Natural source	30	N/A	M = 13	Vit D sufficiency increases the anti-inflammatory cytokine response to muscular injury.
104	Nieman D.C. [132]	2013	Double-blind experimental design	Vit D2	Supplement and natural source	13	Group 1 = 27.3 ± 0.9 Group 2 = 27.1 ± 1.5	N/A	In crew athletes, the intake of 3800 iu/day for 6 weeks of Vit D increased the Vit D level significantly, with no impact on muscle function tests post-eccentric exercise.
105	Barker T. [133]	2013	Randomized, double-blind, placebo-controlled experimental design	Vit D	Supplement	28	Group 1 = 31.0 ± 5 Group 2 = 30.0 ± 6	M = 28	To promote skeletal muscle strength recovery, Vit D supplement intake is recommended, especially after intense exercise in physically active adults.
106	He C.-S. [134]	2013	Cross-sectional	Vit D	Natural source	28	All = 21 ± 3	M = 184 F = 83	A low Vit D level was associated with lower pro-inflammatory cytokine production by monocytes and lymphocytes. Low Vit D levels increase the risk of systemic immunity in endurance athletes.
107	Askari G. [135]	2013	Randomized, placebo-controlled, double-blind clinical trial	Vit C	Supplement	225	M = 21.0 ± 1.6	M = 60	Quercetin and Vit C supplementation may not be beneficial in lipid profile improvement, although it may reduce induced muscle damage and the body fat percentage.
108	Taghiyar M. [136]	2013	Randomized, double-blind clinical trial	Vit C, Vit E	Supplement	60	Group 1 = 31.3 ± 1.8 Group 2 = 38.5 ± 1.6 Group 3 = 33.9 ± 1.5 Group 4 = 38.1 ± 1.4	F = 64	Vits C and E supplementation can be beneficial in reducing muscle damage indices during aerobic exercises.
109	Magee P.J. [137]	2013	Observational study	Vit D	Supplement	64	All ≥ 18	M = 84	Vit D supplement is recommended during the winter and early spring months to overcome insufficiency in athletes.
110	Peeling P. [138]	2013	Cross-sectional	Vit D	Natural source	84	All = 16 ± 4	M = 43 F = 29	Vit D deficiency is common among athletes; hence, coaches are encouraged to recommend a stretching warm-up routine in an outdoor setting during the winter season specifically.
111	Sureda A. [139]	2013	Randomized, double blind clinical trial	Vit C, Vit E	Supplement	72	Group 1 = 32.7 ± 9.2 Group 2 = 36.4 ± 9.7	M = 14	Neutrophils protein oxidation-induced exercise is reduced due to Vit E and Vit C antioxidant supplement intake proteins. Vit E and Vit C intake did not alter the adaptive response of the antioxidant and increased the gene expression of catalase and glutathione peroxidase.
112	Garelnabi M. [140]	2012	Randomized clinical trial	Vit E	Supplement	14	M = 32.4 ± 8.7 F = 34.2 ± 10.0	M = 195 F = 260	Vit E had no impacts on exercise oxidative stress and inflammation.
113	Spradley B.D. [141]	2012	Randomized, double-blind, placebo-controlled cross-over design	B Vits	Supplement	60	M = 28 ± 5	M = 12	The performance reaction and endurance of lower body muscles are enhanced significantly after consuming supplements pre-strenuous exercise. Moreover, spare energy and reduced fatigue delay fatigue.
114	Czaja J. [142]	2011	Clinical trial	Magnesium and B6	Supplement	12			Supplementation food with magnesium is recommended
115	Patlar S. [143]	2011	Clinical trial	Vit E	Supplement	NR	M = 22.1 ± 0.5	M = 7	There is a significant interference of mineral and electrolyte metabolism due to the intake of Vit E in elite athletes.
116	Halliday T.M. [144]	2011	Clinical trial	Vit D	Natural source	7	M = 20.1 ± 1.9 F = 19.9 ± 1.5	M = 18 F = 23	Among university athletes, the intake of Vit D decreases the risk of frequent illness.
117	Abolghasem R. [145]	2011	Clinical trial	Vit C	Supplement	41	N/A	N/A	The level of serum CPK decreased significantly with the Vit C intake.
118	Louis J. [146]	2010	Clinical trial	Vit-Mineral Complexes	Supplement	N/A	Group 1 = 50.8 ± 6.5 Group 2 = 47.7 ± 6.3	N/A	Micronutrient supplements decrease muscle inflammation, which improves strength training.
119	Chatterjee P. [147]	2010	Clinical trial	Vit E	Supplement	20	F = 21–25	F = 25	Among inactive women and during phases of the menstrual cycle, the intake of 400 mg/day for 1 week of a Vit E supplement improved VO2max, maximum voluntary ventilation, oxygen pulse, and endurance capacity. Vit E supplementation should be considered to improve the endurance performance of females.
120	Fage N. [148]	2010	Clinical trial	Vit D	Supplement	25	All = 40.6 ± 12.1	M = 15 F = 5	During anaerobic metabolism, there was no association between Vit D serum status and MAV.
121	Zaǐtseva I.P. [149]	2010	Clinical trial	Vit–Mineral Complexes	Supplement	20	All = 18–22	N/A	A higher dose of minerals decreased the absorption percentage of iron, copper, and magnesium and increased fecal and urinary ME excretion.
122	Dalbo V.J. [150]	2010	Double-blind, randomized crossover design	Mineral Antioxidant Complex (Mac)	Supplement	49	M = 23.6 ± 3.7	M = 15	The performance of aerobic exercise and the lactate response did not differ due to the mineral antioxidants’ complex intake.
123	Karandish M. [151]	2008	Double blind, randomized controlled trial	Vit C	Supplement	15	F = 20–33	F = 49	Among healthy young women who engage in moderate-intensity exercise, the intake of 500 mg/day for two weeks of a Vit C supplement had no impact on oxidative stress markers.
124	Al-Khalidi M.J.M. [152]	2009	Clinical trial	Vit D3	Supplement	219	N/A	F = 18	The biomechanical variables studied helped the researchers to determine the strengths and weaknesses of the sample in the skill of spike. Applications of resistance exercises of all types with the aids with doses of Vit D had the effect of improving the performance of the skill of spike.
125	Sureda A. [153]	2008	Randomized clinical trial	Vit C, Vit E	Supplement	49	M = 32–36	M = 14	A moderate amount of Vits antioxidants supplements reduced oxidative damage due toexercise and lipid peroxidation due to intense exercise and maintained the exercise cellular adaptation.
126	Nakhostin-Roohi B. [154]	2008	Double-blind, placebo-controlled trial	Vit C	Supplement	14	Group 1 = 21.5 ± 0.8 Group 2 = 22.1 ± 0.6	M = 16	The intake of Vit C prevented muscle damage and lipid peroxidation post-endurance exercise but had no impact on inflammatory markers.
127	Cholewa J. [155]	2008	Clinical trial	Vit C	Supplement	16	All = 23.9 ± 2.6	N/A	Among basketball players, the levels of blood antioxidants and VO2max were not affected by the intake of a Vit C supplement.
128	Machefer G. [156]	2007	Cross-sectional	Antioxidant Vits (A-Tocopherol, Vc, Β-Carotene, Retinol)	Natural source	21	M = 41.4 ± 1.8	M = 19	Among ultra-endurance athletes, a low intake of antioxidant Vit led to insufficient energy intake.
129	Disilvestro R.A. [42]	2007	Clinical trial	Vit D, Calcium	Supplement	19	F = 18–24	F = 24	To improve bone health among young adult women, it is recommended to consider exercise along with the intake of micronutrients.
130	Gaeini A.A. [44]	2006	Randomized controlled trial	Vit E	Supplement	24	N/A	M = 20	Student athletes’ performance was not affected by Vit E supplementation.
131	Johnston C.S. [157]	2006	Preliminary study	Vit C	Natural source	20	All = 18–38	N/A	Fatigue might be reflected due to fat oxidation inhibition, which is related to a low Vit C status during submaximal exercise.
132	Davison G. [158]	2006	Clinical trial	Vit C	Supplement	78	M = 26 ± 2	M = 9	Post-endurance exercise neutrophil depression was not prevented by the intake of Vit C supplements when consumed for 2 weeks.
133	Fischer C.P. [159]	2006	Randomized controlled trial	Vit C, Vit E	Supplement	9	Group 1 = 25.6 Group 2 = 22.3 Group 3 = 24.1	M = 21	During an acute exercise program, the intake of Vit E and Vit C for 28 days suppressed the heat shock protein.
134	Machefer G. [160]	2006	Cross-sectional	Vit E, Vit C, Beta Caroten, Retinol	Natural source	21	All = 41.4 ± 1.8	N/A	Athletes suffered from an inadequate intake of antioxidant Vits.
135	Bryer S.C. [161]	2006	Randomized control trial	Vit C	Supplement	19	Group 1 = 24.4 ± 1.7 Group 2 = 21.4 ± 0.8	M = 18	The intake of Vit C before exercise would inhibit muscle stress, postpone creatine kinase release, and suppress the oxidation of bloodglutathione, with a minor impact on the loss of muscle function.
136	Fry A.C. [162]	2006	Randomized control trial	Vits (A, B1, B2, B3, B6, B5, B9, B12, Biotin, C, D, E) Minirals (Calcium, Chromium, Iodine, Iron, Magnesium, Manganese, Potassium Selenium, Sodium, Zinc)	Supplement and natural source	18	Group 1 = 25 ± 4 Group 2 = 23 ± 2	M = 14	The performance post-short-term anaerobic exercise is enhanced by the intake of a micronutrients supplement, but not in well-trained individuals consuming an adequate diet.
137	Senturk U.K. [163]	2005	Clinical trial	Vits (C, E, A)	Supplement	14	Group 1 = 20.6 ± 0.33 Group 2 = 19.8 ± 0.44	M = 18	In relation to a series of exhausting exercises, the risk of hemorheological and exercise-induced death would be prevented by antioxidant Vit treatment.
138	Davison G. [164]	2005	Clinical trial	Vit C	Supplement	18	M = 25 ± 2	M = 6	In prolonged exercise, high doses of Vit C with or without carbohydrates were traced to a minor impact on the hormonal, interleukin-6, or immune response.
139	Herrmann M. [165]	2005	Case-control	Vit B12, Folate	Natural source	6	Group 1 = 38 ± 7 Group 2 = 38 ± 9	N/A	Among recreational athletes, mma (methylmalonic acid) was not associated with Vit B12 metabolism.
140	Rousseau A.-S. [166]	2004	Cross-sectional study	Vits (E, C), B-Carotene, Carotenoids	Natural source	118	All = 26.8 ± 6.8	M = 84 F = 34	Carotenoids as exogenous antioxidants have important protective roles and should be considered by athletes.
141	Cui J.-H. [167]	2004	Randomized control trial	Vit Tablet	Supplement	118	N/A	M = 40	To delay sport fatigue and lipid peroxidation after hypoxia condition exercise, it is recommended to consume acetazolamide, highland-Vit-tablets, and redbull beverages.
142	Viitala P.E. [168]	2004	Clinical trial	Vit E	Supplement	40	Group 1 = 23.3 ± 3.8 Group 2 = 24.2 ± 3.7	M = 15 F = 12	No significant difference between Vit E supplements and placebo in reducing oxidative damage and lipid peroxidation measures between the trained and untrained groups.
143	Thompson D. [169]	2004	Experimental design	Vit C	Supplement	27	Group 1 = 25.3 ± 1.4 Group 2 = 22.6 ± 1.7	M = 14	The intake of a Vit C supplement post-eccentric exercise has no impact on the interleukin-6 level.
144	McAnulty S.R. [170]	2004	Randomized, double-blind, crossover design	Vit C, Polyphenols	Vit C: supplement, polyphenols: natural source	14	M = 23.8 ± 2.5	M = 9	Blueberries supplement increases the level of rooh but not the f2-isoprostane level more than Vit C.
145	Avery N.G. [171]	2003	Clinical trial	Vit E	Supplement	9	N/A	M = 18	The intake of Vit E supplement post-resistance concentric/eccentric exercise did not prevent oxidative stress, membrane damage, and low performance.
146	Bryant R.J. [172]	2003	Clinical trial	Vitc, Vit E	Supplement	18	M = 22.3 ± 2	M = 7	The intake of Vit E (400 IU/day) decreases the tissue damage but not the performance more than Vit C. To prevent the damage effect of exercise, it is recommended that athletes consume Vit E and Vit C via their diet.
147	Tauler P. [173]	2003	Experimental study	Vits (A, B1, B2, B6, B12, C, D, E, Niacin, Folic Acid) and Minerals (Sodium, Potassium, Calcium, Phosphorous, Magnesium, Iron, Zinc, and Iodine).	Vit C: supplement; dietary intake for all micronutrients	7	Group 1 = 25.0 ± 1.5 Group 2 = 24.4 ± 1.1	M = 16	The excessive antioxidant nutrients’ intake, which contains Vit C, increases ascorbate in order to prevent negative impacts on rbc and body tissue due to post-exercise oxidative stress.
148	Schneider M. [174]	2003	Placebo controlled, cross-over	Vit E, Vit C	Supplement and natural source	16	M = 26.5 ± 0.9	M = 13	With moderate oxidative stress, Vit E supplement intake would not be able to enhance the level of Vit C.
149	Sacheck J.M. [175]	2003	Clinical trial	Vit E	Supplement	13	Group 1 = 26.4 ± 3.3 Group 2 = 71.1 ± 4.0	M = 32	Vit E supplement would prevent the oxidative stress due to eccentric exercise.
150	König D. [176]	2003	Cross-sectional	B12, Folate	Blood sample	32	M = 27.1 ± 5.3	M = 39	High leveld of plasma folate among athletes decrease hcy levels due to the highest training volume.
151	Mel’nikov A.A. [177]	2003				39			
152	Thompson D. [178]	2003	Clinical trial	Vit C	Supplement	NR	Group 1 = 23.6 ± 1.4 Group 2 = 24.3 ± 1.7	M = 16	The immediate intake of Vit C post-exercise is not proper for promoting recovery.
153	Tauler P. [179]	2002	Clinical trial	Vit E, Vit C, and Beta-Carotene	Supplement	16	All = 23.3 ± 2.0	N/A	The activity of superoxide dismutase and catalase antioxidant enzymes in neutrophils is enhanced by antioxidant supplementation.
154	Childs A. [180]	2001	Double-blind, placebo-controlled	Vit C	Supplement	20	M = 24.4 6 3.6	M = 14	The immediate intake of Vit C and NAC supplement after injury would have a negative effect on tissue damage and oxidative stress.
155	Krause R. [181]	2001	Clinical trial	Vit C	Supplement	14	M = 29 ± 3	M = 10	Post-strenuous exercise, neutrophil dysfunction is corrected due to the intake of Vit C supplement.
156	Thompson D. [182]	2001	Clinical trial	Vit C	Supplement	10	Group 1 = 23 ± 2 Group 2 = 25 ± 2	M = 16	Long-term Vit C supplement intake would have a reasonable and beneficial impact on the recovery phase due to the unusual exercise protocol.
157	Akova B. [183]	2001	Clinical trial	Vit E	Supplement	16	Group 1 = 26 ± 6 Group 2 = 27 ± 8	F = 18	Oestradiol protects from oxidative injury, more so than Vit E. Both had no impact on exhausted muscle performance.
158	Petersen E. [184]	2001	Clinical trial	Vit C, Vit E	Supplement	18	Group 1 = 28 Group 2 = 26	M = 20	The level of Vit C and Vit E is raised significantly due to the supplements intake, although its level does not affect the immune indicators of cytokine, lymphocyte responses, or muscle enzymes due to exercise.
159	Sacheck J.M. [185]	2000	Experimental	Vit E, Vit C, Beta Carotene	Natural source	20	F = 18–25	F = 22	Vit E intake, even not from supplements, would be sufficient to protect from the oxidative stress which is yielded due to the moderate-intensity exercise.
160	Kawai Y. [186]	2000	Clinical trial	Vit E	Supplement	22	F = 21.2 ± 0.5	F = 10	The priority of Vit E intake is RBC protection against oxidative damage. The sufficiency level of serum Vit E promotes its shifting from the serum to a steady RBC-α-tocopherol level due to exercise.
161	Chung T.-W. [187]	2000	Clinical trial	Vit C	Supplement from fruit	10	N/A	M = 20	A minor increase in the antioxidant level when athletes consume 500 mg of Vit C from fruits during moderate/high-intensity endurance training; however, it did not balance the oxidative stress. Athletes who engage in long-term endurance-training would benefit from Vit C extracted from fruit.
162	Beshgetoor D. [188]	2000	Prospective, observational study	Calcium	Natural source	20	F = 49.6 ± 7.9	F = 30	No significant interaction effect of the sport and dietary calcium intake was noted for bmd at any site.
163	Monnat A. [189]	2000		N/A	N/A	30	N/A	N/A	N/A
164	Sürmen-Gür E. [190]	1999	Clinical trial	Vit E	Supplement	N/A	M = 12–24	M = 36	Vit E supplementation had an insufficient impact on plasma lipid peroxidation after exercise.
165	Krumbach C.J. [191]	1999	Cross-sectional	Multi-Vits + Minerals	Supplement and natural source	36	All ≥ 19	M = 266 F = 145	Calcium and iron supplementation are commonly consumed by females, while males consume Vit B12 and Vit A supplements. There were some variations in the Vit/mineral supplement habits of the athletes by gender, ethnicity, and sport.
166	Virk R.S. [192]	1999	Nonrandomized clinical trial	Vit B6	Supplement	411	N/A	M = 11	Through intensive endurance exercise, Vit B6 supplementation can alter plasma FFA and amino acid levels.
167	Savino F. [193]	1999	Comparative study	Vits	Supplement	11	Group 1 = 6–12 Group 2 = 9	N/A	Subjects who are at risk of Vit deficiency or highly demanding would benefit from supplements intake.
168	Rourke K.M. [194]	1998	Double-blind clinical trial	Calcium	Supplement	40	F= 18–22 Year	F = 30	Higher calcium intakes promote some benefit to bone mineral density.
169	Alessio H.M. [195]	1997	Clinical trial	Vit C	Supplement	30	M = 33.0 ± 2.6	M = 9	Vit C supplementation for 1 day/2 weeks reduces the oxidative stress induced by exercise.
170	Oostenbrug G.S. [196]	1997	Double-blind randomized	Vit E	Supplement	9	M = 19–42	M = 24	Endurance performance is not improved by fish oil supplements. Due to the increase in oxidative stress post-endurance exercise, fish oil may act as an antioxidant.
171	Hartmann A. [197]	1995	Clinical trial	Vit E	Supplement	10	M = 29–34	M = 8	Vit E prevents exercise-induced DNA damage and indicates that dna breakage occurs in WBC after exhaustive exercise as a consequence of oxidative stress
172	Sobal J. [198]	1994	Cross-sectional	MultiVites + Minerals	Supplement and natural source	8	N/A	N/A	Supplement use by these adolescents appears to be motivated more by health reasons than by sports performance. It is suggested that it may be useful to assess Vit/mineral supplement use by adolescents and to provide education and counseling about diet, nutrition, and exercise for those who use them as ergogenic aids to improve athletic performance.
173	Men’shikov I.V. [199]	1994	Comparative study	Vit E	Supplement	742	N/A	N/A	The use of Vit E resulted in a decrease in the energy value of exercises performed by the athletes under normo- and hyperthermic conditions as well as changes in blood and erythrocyte membrane lipid composition and blood calcium ion concentration.
174	Rokitzki L. [200]	1994	Cross-section	Vit B6	Supplement	N/A	Bodybuilding = 25.3 ± 6.4 Wrestling = 20.6 ± 2.7 Basketball = 26.1 ± 4.7 Soccer = 23.5 + 2.8 Handball = 23.8 ± 7.9	M = 45 F = 12	Athletes’ Vit B6 level is not yet assessed due to the absence of generally valid reference values.
175	Rokitzki L. [201]	1994	Clinical trial	Vit B6	Supplement	57	M = 35.6 ± 9.8	M = 13	With a balanced diet, the intake of exogenous Vit B6 is not necessary.
176	Rokitzki L. [202]	1994	Cross-section	ViteB2	Supplement	13	N/A	Athletes Group: M = 50 F = 12	B2 Vit is favored among performance athletes.
177	Lorino A.M. [203]	1994	Single-blind crossover	Vit E	Supplement	78	M = 22.0 ± 1.0	M = 7	Vit E intake did not reduce the lung clearance exercise-induced increase.
178	Jakemanl P. [204]	1993	Double-blind clinical trial	Vit E and Vit C	Supplement	7	All = 19.6	M = 16 F = 8	There was a reduction in the loss of contractile function post-eccentric exercise and in the first 24 h of recovery in the group supplemented with Vit C but not Vit E. Prior supplementation with Vit C can attenuate eccentric exercise-induced muscle damage. It is proposed that the effect of Vit C supplementation on contractile function, particularly LFF, could be to protect vital cell structures such as the SR from oxidative stress and free radical injury.
179	Nasolodin V.V. [205]	1993	N/A	Vit C, Vit P, and Vit Complex (Ascorutine, Thiamine, Riboflavin, Pyridoxine, Cyan Cobalamin, Folic Acid)	N/A	24	N/A	N/A	The metabolism of iron, copper, and manganese is affected by the intake of Vit c or Vit complex (ascorutine, thiamine, riboflavin, pyridoxine, cyan cobalamin, folic acid).
180	Bazzarre T.L. [206]	1993	Cross-sectional	Vit-Mineral Supplement	Supplement	N/A	N/A	N/A	Subjects who intake supplements might reflect healthy lifestyle practices.
181	Klausen T. [207]	1993	Cross-sectional	Vit D and Calcium	Natural source	91	M = 41–50	M = 9	Endurance training impacts the plasma Vit D level and pth.
182	Fogelholm M. [208]	1993	Clinical trial	B-Complex	Supplement	9	All = 18–32	M = 24 F = 18	Activation coefficients acs (Vit B1, B2, and B6) were not associated with blood lactate.
183	Maxwell S.R.J. [209]	1993	Clinical trial	Vit E and Vit C	Supplement	42	All = 19.6 ± 0.3	M = 16 F = 8	Vits supplementation enhanced the plasma capacity of antioxidants due to one hour of eccentric exercise.
184	Meydani M. [210]	1993	Double-blind clinical trial	Vit E	Supplement	24	Young Group = 22–29 Adult Group = 55–74	M = 21	Vit E dietary supplementation for 48 days diminished free radical-mediated exercise oxidative damage, reduced oxidative injury, and increased muscle a-tocopherol.
185	Mikalauskaǐte D.A. [211]	1992				21			
186	MEYDANI M. [212]	1992				NR			
187	Telford R.D. [213]	1992	Clinical trial	Vits (B1, B2, B6, C, E, A, B12, Folate) and Six Minerals (Cu, Mg, Zn, Ca, P, Al)	Supplement	NR	M = 17.3 ± 1.4 F = 17.3 ± 1.1	M = 50 F = 36	Micronutrients supplementation for 7–8 months promotes blood Vits levels but not mineral levels, and some blood nutritional indicators may vary according to sex (the values generally being higher in females, with significant differences for Vits B2, C, and E and copper and aluminum).
188	Fogelholm M. [214]	1992	Cross-sectional	Vits (B1, B2, B6), Magnesium, Iron, and Zinc	Natural source	86	Group 1 = 24.0 ± 0.6 Group 2 = 26.0 ± 0.6	F = 39	A protocol of 24-week fitness exercise effectively increased VO2max, while it did not affect thiamin, riboflavin, magnesium, iron, and zinc status.
189	Deuster P.A. [215]	1991				39			
190	Colgan M. [216]	1991	Double-blind crossover trial	All Vits and Minerals	Supplement	NR	N/A	M = 12 F = 11	Among endurance intense training athletes, RDA is enough, along with iron supplements.
191	Pieralisi G. [217]	1991	Double-blind, randomized, crossover		Supplement	23	M = 21–47	M = 50	Ginseng preparation improved muscular oxygen utilization and hence enhanced work capacity.
192	Miric M. [218]	1991				50			
193	Cannon J.G. [219]	1991	Double-blind placebo-controlled protocol	Vit E	Supplement	NR	Group 1 = 25 ± 3 Group 2 = 65 ± 2	M = 21	Vit E supplement significantly affected IL-1 and IL-6 production, with no significant impact on exercise-related changes
194	Faber M. [220]	1991	Cross-sectional	Calcium, Iron, Magnesium, Phosphorus, Vit A, Thiamin, Riboflavin, Nicotinic Acid, Vit B6, Folic Acid, Vit B12, Ascorbic Acid	Natural source	21	M = 22.1 ± 3.8 F = 22.3 ± 2.9	M = 20 F = 10	With a sufficient energy intake, males consumed an adequate amount of all micronutrients, while females digested insufficiently for calcium, iron, and magnesium.
195	Cannon J.G. [221]	1990	Clinical trial	Vit E	Supplement	30	Group 1 = 22–29 Group 2 = 55–74	M = 21	Vit E supplementation had a positive impact on damaged tissue, as it promotes the accumulation of neutrophil.
196	Van Erp-Baart A.M.J. [222]	1989	Survey research	Calcum, Iron, Vites (C, A, B1, B2, B6)	Natural source and supplement	Nr	N/A	N/A	A sufficient intake of Vit and minerals was notice when the energy intake ranged between 10 and 20 mj/day. There was a positive correlation between calcium and iron intake and energy intake. Hence, low energy intakes challenge ca and iron.
197	Satoshi S. [223]	1989	Experimental trial	Vit E	Supplement	N/A	M = 20.3 ± 0.3	M = 21	Post-exercise Vit E supplementation significantly reduces the malondialdehyde level, the leakage of enzymes, and lipid peroxidation.
198	Guilland J.-C. [224]	1989	Clinical trial	Vit B1, B2, B6, C, A, and E	Natural source and supplement	21	Group 1 = 19.6 ± 0.56 Group 2 = 22.5 ± 0.4	M = 55	Daily Vits supplements for one month significantly improve serum Vit C, plasma plp, and erythrocyte tpp concentrations, etk and egr basal activities, east basal activities (only in young athletes), and etk, egr, and east activation values.
199	Weight L.M. [225]	1988	Double-blind, placebo-controlled crossover study	Vits (E, D, C, A, B1, B2, B6) and Minerals (Selenium, Iodine, Phosphorus, Ca, Iron, Zn, Cu, K, Mg)	Supplement	55	M = 20–45	M = 30	A normal diet intake is enough, and there is no need for supplementation.
200	Weight L.M. [225]	1988	Placebo-controlled crossover	Multi-Vits and Minerals	Supplement	30	M = 31.9 ± 10.6	M = 30	
201	Manore M.M. [226]	1988	Clinical trial	B6	Natural source	30	Group 1 = 25.6 ± 4 Group 2 = 24.4 ± 3.2 Group 3 = 55.8 ± 4.8	F = 15	Among females, the intake of Vit B6 may alter serum fuel substrates during exercise, depending on her age.
202	Bell N.H. [227]	1988	Clinical trial	Calcium, Phosphorus, Sodium, Potassium, Magnesium	Natural source	15	Group 1 = 19–36 Group 2 = 26 ± 1	M = 28	
203	Klepping J. [228]	1988	Clinical trial	Minerals (P, Mg, Ca, Fe) and Vits (C, B1, B2, B6)	Natural source	NR	Group 1 = 15.16 ± 1.6 Group 2 = 23.91 ± 4.85	M = 265	Even with a high energy intake, deficient micronutrient intakes were common among many athletes.
204	Manore M.M. [229]	1987	Comparative study	B6	Natural source	N/A	Group 1 = 25.6 ± 4 Group 2 = 24.4 ± 3.2 Group 3 = 55.8 ± 4.8	F = 15	Vit B6 metabolism did not change due to exercise, nor training.
205	Zuliani U. [230]	1985	N/A	Mineral Salts	N/A	NR	N/A	N/A	N/A
206	Butturini U. [231]	1984				N/A			
207	Walter M.C. [232]	1984				NR			
208	Grandjean A.C. [12]	1983	N/A	N/A	N/A	NR	N/A	N/A	N/A
209	Rusin V.I. [233]	1982				N/A			
210	Borisov I.M. [234]	1980	Comparative study	Vit C, Vit P	Supplement	NR	N/A	N/A	N/A
211	Helgheim I. [235]	1979	Double-blind experimental design	Vit E	Supplement	1203	All = 19–24	M = 2 F = 24	Post-exercise intake of Vit E had no impact on serum enzyme levels.
212	Leklem J.E. [236]	1979				26			
213	Laricheva K.A. [237]	1979	N/A	Vits (A, B1, B2, B6, Pp, C)	N/A	NR	N/A	N/A	N/A
214	Dam B.V. [238]	1978	Double-blind test	B1, B2, and B6	Supplement (granulated multi-Vit electrolyte preparation)	N/A	F = 18.3 ± 3.0 M = 18.9 ± 3.5	M = 33 F = 7	B Vits deficiency is noticed due to the high rate of energy metabolism, the high body core temperature, and sweat loss.
215	Haralambie G. [239]	1976	Clinical trial	Vit B	Supplement	40	M = 17–38	M = 25	An intake of 10 mg of riboflavin lowers neuromuscular irritability.
216	Shephard R.J. [240]	1974	Matched-pair trial under near-double-blind conditions	Vit E	Supplement	25	M = 20.3 ± 1.6	M = 16	Vit E had no advantage for swimmers.
217	Bailey D.A. [241]	1970	Double-blind	Vit C	Supplement	NR	M = 24.5 ± 3.5	M = 40	There was no significant impact of Vit C on respiratory and oxygen utilization before, during, and after exercise in smoking and nonsmoking subjects.

## Data Availability

All data are available in the manuscript.

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
