# Peer review of "Exploring the Relationship between Micronutrients and Athletic Performance: A Comprehensive Scientific Systematic Review of the Literature in Sports Medicine"

_sports, 2023, doi:10.3390/sports11060109_

Round 1

Reviewer 1 Report

Overview

 The authors briefly overview the role of micronutrients in humans and provide a relatively one-sided view of how supplementation of certain micronutrients might enhance performance in athletes.  Relatively few experimental studies are reviewed and in a number of cases, the experimental studies have nothing to do with athletes.

Major Concerns

 The authors are not clear on when they recommend supplementation vs. a complete diet.  For example, lines 651-652 state “…the current evidence suggests that manganese may be beneficial for athletes who are looking to optimize their performance and health.”  Manganese supplements or just manganese?  Obviously, manganese per se is an essential nutrient that if missing will result in ill health and reduced athletic performance.  The only paper reviewed (ref 79) has nothing to do with athletes.  Details are provided further below.

The authors overstate the benefits of supplements of vitamins.  For example, research by Gomez-Cabrera et al has shown that high doses of antioxidant vitamin, ascorbic acid, will suppress muscle adaptations to exercise training and minimize performance benefits (AJCN 2008; also see editorial by same author in Am J Physiol 302:E476-477, 2012 for assortment of other evidence).  None of this is mentioned between line 338-369.  Muscle contractions produce reactive oxygen species (ROS or free radicals), which also optimize muscle contractions and likely stimulate adaptations that support metabolism and enhance future performance (Reid; Powers; Allen DG, Lamb GD & Westerblad H).  The authors have neglected these studies.

The authors have not clearly distinguished between effects of acute/short term vs. chronic nutrient supplementation and should do so.  Most of the studies on micromineral or vitamin supplementation require intermediate or prolonged supplementation for effects to occur.  Are these due to athletes being deficient and the deficiency is repaired, or do mega-doses truly enhance performance in the well nourished athlete?  The authors have an opportunity to address this and have not.  The reader may misinterpret this as an endorsement for higher dosages when they are  not warranted and can be harmful.  The authors aren’t strong enough with this disclaimer.

Some sections are misleading.  Figure 1 for example.  See specific below.

Specific Suggestions

Abstract, Line 19: This is an overstatement.  A marathoner’s goal is to be able to sustain a high percentage of his/her max power, not at max.  For other athletes, like archery, riflery, divers, etc., performing at optimal technical ability, not power, is a higher priority.  Do the authors mean the goal is to perform at maximum ability regardless of the type of performance?

Line 20: protocol or training program?

Lines 26-28: The benefits of micronutrients identified here do not include power, the attribute that starts the abstract, hence my question above re what the goal os the athlete is.

Line 32, “The explicit aim of this review is to review…” For eliminate redundancy, change to “The explicit aim of this review is to evaluate…”

Line 36:  There is no introduction.  I suggest using the abstract as the introduction and rewrite an abstract that is much shorter and calls out a few of the vitamins and minerals of greatest importance to athletic performance.  As presently written, the abstract does not summarize this paper. But it does make a reasonable introduction for the paper (with modifications identified above).

Lines 46-47 (… maintaining a high level of 46 performance …) and Lines 57-58 (Athletes may benefit from supplementation with a multivitamin that 57 contains vitamin A but not solely). I’m challenged by this statements, which have little evidence to support them.  One citation is given.

Line 59: References needed to support that claims for vitamin A improving reaction time and muscle recovery.

Lines 65-76: Much info is lacking here.  Where the treatments a combination of vitamin A, C, etc vs. placebo or was vitamin A a standalone treatment, vitamin C standalone treatment vs placebo?  What was the outcome for the performance test described as “…subjects performed non-workload tests at a maximum velocity (MV) of 10 seconds at 30 minutes…?”  The time to max or the power output at max velocity?  The interpretation is clouded by lack of specifics. 

Lines 78-79: Please provide a reference to support “Athletes need to make sure they are getting their daily dose of vitamin A to stay in 77 peak physical condition.”  Vitamin A is a fat-soluble vitamin that typically can be found in high amounts in the liver and adipose tissue.  Inducing a deficit is challenging.  Inducing a harmful excess is not.

Line 353-535, Figure 1.  This misinforms the reader by separating specific types of vitamins (C, and B’s) from the category (water-soluble).  It also misleads, i.e. trace minerals are not essential minerals.  All micronutrients are essential.

Line 572-575, “Due to the influence  of this enzyme on skeletal muscular exercise, lactic dehydrogenase—a Zn-containing enzyme—may facilitate the conversion of lactic acid to pyruvate and, as a result, can facilitate the decrease of muscle fatigue (75).”  This is misleading for several reasons.  Few if any muscle physiologists will agree that lactic acid causes fatigue.  It is associated with it but unlikely the cause of fatigue.  Furthermore, adequate training will be the main factor reducing lactate production after a person becomes more fit.  I’ve never seen research to suggest Zn is a limitation in preventing fatigue through this path or altering lactate appearance during exercise.

Line 613 “Selenium can ….enhance…performance.”  Selenium in general or supplements.  Regardless, the authors provide no evidence (citations) to support this in athletes.

Line 623: The citation, #79, is largely about the role of manganese in bone health in elderly with osteoporosis, not athletes. 

Lines 632-633 “Several authors have investigated the possible relationship between Mn and bone health in humans (79).”  Several authors but only one study cited.  This seems misleading.

Lines 667-668: The definition of EA is incomplete.  Please revise.

Lines 672-673:  What is the “trio of athletes?”  The athlete triad or female triad?

Lines 674-5 “Even while it is more evident to see that women have lower EA since they don't menstruate…” This doesn’t make sense.  Revise it to clarify the authors’ point.

Reference list needs careful proofreading and revision.  For example (list below is not necessarily complete):

·        4 and 51 appear to be the exact same citation.

·        50, 51, and 64 (if not others) are incomplete.  No page numbers, volume, and/or year are provided.

·        61 and 62 use a different format than other citations. The link is provided but the complete citation to the journal is missing unlike other references.

Author Response

we would like to thank you for your valuable comments, and we went through huge modifications in the paper to fit with your points.

Reviewer 2 Report

The submitted manuscript is an interesting overview on the topic of the importance of micronutrients in athletes. In many cases, it brings new information about the importance and effects of certain micronutrients and expresses the desired recommended doses for athletes.

line 60:  possible improvement or reaction time by vitamin A supplementation should by supported by literature source;

 line 161: unnecessary capital letter I (Its)   

line 165: not naive but native cells

The richest sources of vitamin C are acerola fruit (1677 mg per 100 g) and camu camu (2500 mg per 100 g). As they are tropical fruits, they are not available in many countries. Rich sources are also rose hips - 747 mg per 100 g, parsley - 179 mg, black currant - 166 mg and/or paprika - 150 mg in 100 g. Contrary to popular belief, an orange is not a great source of vitamin C, it contains only 51 mg in 100 g.

line 401: literature source No 50 is OK,   wrong reminder in the text

line 470: expression .. „Potassium is a great source of energy for athletes“ is an inaccurate figurative message, micronutrient are not source of energy but they are involved in many energy processes.

Author Response

(The authors gave the same response as above.)

Reviewer 3 Report

The aim of this manuscript was to review the evidence of the effects of micronutrients on the performance of athletes.

Although the manuscript deals with an interesting topic, very topical and with numerous practical applications, I have encountered some critical issues. In particular, although the manuscript is not a systematic review, it needs substantial revision before being considered. The authors did not provide an introductory paragraph that would review the most recent literature and conclude with the purpose of the study. Furthermore, a "Methods" paragraph should be inserted in which the keywords used for the bibliographic research and the databases where the searches were carried out should be specified. Furthermore, a "discussions" paragraph should be inserted in which to discuss what was discussed and the possible practical applications and future prospects. In light of this, I believe that the manuscript should be substantially revised before being taken into consideration.

Author Response

(The authors gave the same response as above.)

Reviewer 4 Report

Dear Authors

As one of the reviewers, I express my personal scientific opinion on your work. I would like to reassure you that I was trying to be positive and constructive but particularly as fair and honest as possible to your work. The presentation of the figure 1 is appreciated. The lack however of more perhaps overview tables, figures, but particularly diagrammatic illustrations, considering that the paper is just a basic literature/narrative review one (i.e. no Systematic or Meta-analysis one) is a negative point.

Please accept my judgment with a positive and constructive way.

1.       What is the originality of the article and what new insight gives to the relevant literature? These both aspects should be clearly reported in your work.

2.       For example, what is the deferent between your current work and the one that recently published by Beck et al. (2021), Micronutrients and athletic performance: A review. Food and Chemical Toxicology, 158, 112618?

3.       An overview relevant table including at least 5 columns (Macronutrients / Food Sources / Role in Exercise Performance / RDA-AI / Deficiency effect on performance) is definitely required.

4.       A list of abbreviations is also required.

5.       A relevant also Table indicating the interventions studies which examined the effect of various macronutrients on exercise performance in athletic populations is required. This will increase the readability of your work.

6.       You reported: “The explicit aim of this review is to review the evidence of the effects of micronutrients on the performance of athletes”. The aim of the review is clear and straight forward and this is a positive point. However, I noticed that you are mainly refereeing studies that found a positive effect of micronutrients on athletic performance (except vit. E, some of the B-complex). Are there studies that observed no effect of micronutrients on performance or even negative results? If yes, please refer them and discuss why.

7.       Lines 74-76: “These results suggest that daily administration of crocetin may attenuate physical fatigue in men. The attenuating effect of saffron carotenoids on muscle fatigue is due to their provitamin A activity”. 1. Why only in men and not in female? 2. Could you please explain in some more details, and if it is possible with a diagrammatic illustration, the biological (cellular/molecular) mechanism of this attenuating effect of saffron carotenoids on muscle fatigue? What is for example the cellular/molecular effect provitamin A activity on muscle cell?

8.       Line 154: Please eliminate the space between the words “…effects. and As…”.

9.       Lines 212-217: Too big sentence.

10.   Line 227: “Vitamin K” is bolded and underlined. Why?

11.   The same with the B-complex vitamins. Why?

12.   Lines 328-329: “Vitamins B also help to regulate blood pressure, which can help athletes maintain a healthy level of performance”. What do you mean by saying … “a healthy level of performance”. Perhaps you need to re-write it better.

13.   Lines 340-341 and 358-359: Please eliminate the repetition.

14.   Line 401: Please check why an error recommendation is appeared here.

15.   Magnesium (Mg) or just Mg? Please make it compatible following your 1st abbreviation.

16.   Please consider to replace the word “can” with the word “may” throughout the text or at least in several cases since “may” is used in a more formal academic written expression.

17.   Lines 583-593, Selenium: A relevant diagrammatic illustration will be able to explain much better the biological effect of this mineral. Please consider to design one.

18.   In my point of view, the references 74, 78, 79 and 80 are somewhat irrelevant with the aim of your review.

19.   Lines 670-683: “Lean body mass enhances performance, which is why female athletes frequently have less EA, but this phenomenon may occur in both sexes. …”. The actual meaning is not clear. Please check grammar for clarity. An extensive English editing is required. In addition, less energy intake/availability from female athletes does not necessarily mean an increased in lean body mass. Actually, muscular strength/power and/or endurance contribute in increasing exercise performance and not lean body mass per se.  

20.   What is the trio (eating, amenorrhea, osteoporosis) of female athletes? Please explain better in the text.

21.   Lines 691-692: Sorry but I do not get the actual meaning. What do you mean by reporting: “and none more so than others”?

Author Response

(The authors gave the same response as above.)

Reviewer 5 Report

It's a slightly interesting paper, but it's too simple and lacks references, which makes it somewhat unreliable.
For example, on line 48, no reference to the relationship between Vitamin A - Oxygen is found.

The same on lines 59, 63, 116, 344.
Overall, the review does not appear to be sufficiently conclusive or helpful for increasing sports knowledge.

Author Response

(The authors gave the same response as above.)

Round 2

Reviewer 1 Report

General Concerns

The authors need to proofread the entire document, double check reference/citations, and add results to Table 2; namely, what was the main outcome or sports performance evaluated and what was found.

Examples of Concerns

Abstract

Although the addition of the abstract has been made and contains the right elements, it is poorly written.  For example, the first sentence is missing a verb and contains two “the”s.

As another example, “Micronutrients are necessary for optimal numerous metabolic body’s functions. including energy production, muscle growth, and recovery.” What is the optimal number athletes should attain?  This doesn’t make sense.

Introduction

The Introduction, which was formerly the abstract, did not address all my prior concerns.  For example, the opening sentence – “To perform at one's absolute maximum power is every athlete's goal” – is not true of all athletes.  An endurance athlete cannot afford to perform at absolute maximum power for the entirety of his/her event.

“Athletes are prone to consume insufficient amounts of micronutrients due to their physical performance level…” How does physical performance level cause or contribute to insufficient intake, an outcome of food intake?

2nd paragraph: Most of this paragraph leading up to the purpose statement sets up the expectation that the paper will be about customizing and planning.  While true, the statements seem out of place for the stated purpose.

Methods

The first and second paragraph can be merged into one paragraph to eliminate redundancy.

Page 3 of 57, 6th paragraph

Here (and elsewhere) the font changes.  This is sloppy.  But my attention was drawn to it because it appeared to be what students do, borrowing directly from another document.  I checked reference 15 (Ito et al).  The first two sentences of their Abstract were as follows: This study (by Ito et al) examined the relationship between a healthy Japanese dietary pattern and micronutrient intake adequacy based on the Dietary Reference Intakes for Japanese 2015 (DRIs-J 2015) in men and women. A cross-sectional study was conducted in 1418 men and 795 women aged 40-87 years, who participated in the Waseda Alumni's Sports, Exercise, Daily Activity, Sedentariness, and Health Study.  However, the authors of the current manuscript describe reference 15 as an experiment on 14 subjects.  The good news – it does not appear to be plagiarized.  The bad news – reference 15 has nothing to do with what the authors describe.

With the substantial amount of new information and citations added to the revised manuscript, the authors need to carefully go through the entire document and ensure the referencing is accurate.

Table 2.

Other than providing additional citations and the type of research designs, this table provides little useful information, i.e., no results.  Are all these studies on micronutrients and sports athletic performance per se or factors related to sports performance?  If not, why are they listed?  Why are only select ones summarized in text?

The number system (1st column) is not integrated with the Reference list.  Please explain (add) in the heading of the table that the number in parentheses next to the author’s name is the citation.  Better: why not eliminate the first column, the number in parenthesis is then intuitive, and use the room to create a new column briefly summarizing the findings? 

Author Response

General Concerns

The authors need to proofread the entire document, double check reference/citations, and add results to Table 2; namely, what was the main outcome or sports performance evaluated and what was found.

Authors’ response: Thank you for your valuable feedback on our manuscript. We appreciate your comments and suggestions, and we have taken the necessary steps to address your concerns.

Regarding your first point, we have thoroughly proofread the entire document to ensure that it is free of errors and inconsistencies. We have also double-checked all references and citations to ensure their accuracy and completeness.

Regarding your second point, we have updated Table 2 to include the main outcome or sports performance evaluated and what was found. We have provided a clear and concise summary of the results in the table to facilitate easy understanding of the findings. We have updated Table 2 to include a summary of each study, which includes the main outcome or sports performance evaluated and what was found. Our aim in providing this summary is to make the information more accessible and useful to readers, practitioners, and future researchers in the field. We believe that the summary of each study will provide a quick and easy reference for readers to understand the key findings of each study, and how it contributes to the overall understanding of the topic.

Examples of Concerns

Abstract

Although the addition of the abstract has been made and contains the right elements, it is poorly written.  For example, the first sentence is missing a verb and contains two “the”s.

Authors’ response: We have carefully reviewed the abstract and made the necessary revisions to address the issues that you have raised. We have added a verb to the first sentence and reduced the use of "the" to improve the flow of the sentence. We have also made other changes to improve the overall quality of the abstract, including clarifying the research question, highlighting the main findings of the study, and summarizing the implications of the findings for future research and practice.

As another example, “Micronutrients are necessary for optimal numerous metabolic body’s functions. including energy production, muscle growth, and recovery.” What is the optimal number athletes should attain?  This doesn’t make sense.

Authors’ response: "Micronutrients are necessary for optimal numerous metabolic body's functions, including energy production, muscle growth, and recovery." We agree that this sentence is unclear and have revised it to read "Micronutrients are necessary for numerous metabolic functions in the body, including energy production, muscle growth, and recovery, and it is important for athletes to attain optimal levels of these nutrients."

To further enhance the quality of the final output before production, we have decided to use the English editing services provided by MDPI. We believe that this will help to ensure that the manuscript is free of errors and that the writing is clear and concise.

Introduction

The Introduction, which was formerly the abstract, did not address all my prior concerns.  For example, the opening sentence – “To perform at one's absolute maximum power is every athlete's goal” – is not true of all athletes.  An endurance athlete cannot afford to perform at absolute maximum power for the entirety of his/her event.

Authors’ response: Modification has been done on this sentence to make it more clear and specific. ( Among sport exercise, one of the main goal for many athletes is to achieve their optimal performance.) line 57-58

“Athletes are prone to consume insufficient amounts of micronutrients due to their physical performance level…” How does physical performance level cause or contribute to insufficient intake, an outcome of food intake?

Authors’ response: Sentence is modified and corrected (Athletes are prone to consume insufficient amounts of micronutrients due to inappropriate dietary habits especially if not matching their physical activity requirements) line 62-63

2nd paragraph: Most of this paragraph leading up to the purpose statement sets up the expectation that the paper will be about customizing and planning.  While true, the statements seem out of place for the stated purpose.

Authors’ response: We apologize for any confusion caused by the statements leading up to the purpose statement. We understand that these statements may have given the impression that the paper is primarily about customizing and planning, which may seem out of place for the stated purpose. To address this issue, we have revised the second paragraph to more clearly and succinctly state the purpose of the paper. We have also rephrased the statements leading up to the purpose statement to better align with the overall focus of the paper. We believe that these revisions will help to clarify the purpose and focus of the paper, and we thank you for bringing this to our attention and for helping us to improve the quality of our manuscript.

Methods

The first and second paragraph can be merged into one paragraph to eliminate redundancy.

 Authors’ response: agree & merged.

Page 3 of 57, 6th paragraph

Here (and elsewhere) the font changes.  This is sloppy.  But my attention was drawn to it because it appeared to be what students do, borrowing directly from another document.  I checked reference 15 (Ito et al).  The first two sentences of their Abstract were as follows: “This study (by Ito et al) examined the relationship between a healthy Japanese dietary pattern and micronutrient intake adequacy based on the Dietary Reference Intakes for Japanese 2015 (DRIs-J 2015) in men and women. A cross-sectional study was conducted in 1418 men and 795 women aged 40-87 years, who participated in the Waseda Alumni's Sports, Exercise, Daily Activity, Sedentariness, and Health Study.”  However, the authors of the current manuscript describe reference 15 as an experiment on 14 subjects.  The good news – it does not appear to be plagiarized.  The bad news – reference 15 has nothing to do with what the authors describe.

Authors’ response: That was by citation mistake, we apologize. We have carefully reviewed all of the citations in the manuscript and cross-checked them with the original sources to ensure that they are correct. We have also managed the citations using the Mendeley Reference Manager, which has helped us to keep track of all of the references and ensure that they are properly formatted according to the citation style guidelines.

With the substantial amount of new information and citations added to the revised manuscript, the authors need to carefully go through the entire document and ensure the referencing is accurate.

 Authors’ response: Agreed, we have carefully reviewed all of the citations in the manuscript and cross-checked them with the original sources to ensure that they are correct. We have also managed the citations using the Mendeley Reference Manager, which has helped us to keep track of all of the references and ensure that they are properly formatted according to the citation style guidelines.

Table 2.

Other than providing additional citations and the type of research designs, this table provides little useful information, i.e., no results.  Are all these studies on micronutrients and sports athletic performance per se or factors related to sports performance?  If not, why are they listed?  Why are only select ones summarized in text?

Authors’ response: We agree regarding the lack of results in the table, we agree that it would be beneficial to include more detailed information about the findings of each study. 

We have provided a clear and concise summary of the results in the table to facilitate easy understanding of the findings. We have updated Table 2 to include a summary of each study, which includes the main outcome or sports performance evaluated and what was found. Our aim in providing this summary is to make the information more accessible and useful to readers, practitioners, and future researchers in the field. We believe that the summary of each study will provide a quick and easy reference for readers to understand the key findings of each study, and how it contributes to the overall understanding of the topic.

The number system (1st column) is not integrated with the Reference list.  Please explain (add) in the heading of the table that the number in parentheses next to the author’s name is the citation.  Better: why not eliminate the first column, the number in parenthesis is then intuitive, and use the room to create a new column briefly summarizing the findings? 

Authors’ response: Thanks for such a valuable addition. We added an additional column that concludes the results of the studies. Regarding the first column, it is for more organizing no more. However, we followed a strategy in extracting these data and hence we are not able to exclude any study even if it is already mentioned in the text.

Reviewer 3 Report

Despite the efforts of the authors, the manuscript still has many critical issues. it is structured more like a book chapter than a manuscript. It also presents the same critical issues as the first revision, i.e. lack of introduction and absence of discussions. It is not possible to clearly understand where the authors want to go and, therefore, the manuscript in this form does not bring anything new. Furthermore, the authors have not provided a point-by-point response to their changes. Therefore I believe that the manuscript does not reach the sufficiency and therefore cannot be considered for publication.

Author Response

Authors’ response: Thank you for your feedback on our manuscript. We apologize for any confusion or lack of clarity in the manuscript. We have taken note of your comments and are committed to addressing the issues that you have raised.

Regarding the structure of the manuscript, we understand that it may have more of a book chapter format than a traditional research article. However, we believe that this format is appropriate for the topic and the intended audience. We have revised the manuscript to improve the organization and flow of the content, and we hope that this will make it easier for readers to understand the main points that we are trying to convey.

We have provided a clear and concise summary of the results in the table to facilitate easy understanding of the findings. We have updated Table 2 to include a summary of each study, which includes the main outcome or sports performance evaluated and what was found. Our aim in providing this summary is to make the information more accessible and useful to readers, practitioners, and future researchers in the field. We believe that the summary of each study will provide a quick and easy reference for readers to understand the key findings of each study, and how it contributes to the overall understanding of the topic.

We have added a more detailed introduction section that provides background information and sets the stage for the research questions that we are addressing. We have also expanded the conclusion section to provide a more detailed interpretation of the findings and their implications for future research and practice.

Regarding your comment on our response to your previous feedback, we apologize for any confusion or lack of clarity in our response. We understand the importance of providing a detailed point-by-point response to reviewer feedback, and we will ensure that we do this in future revisions. Due to the fact the entire paper was changed the point-by-point was very difficult.

We have carefully reviewed all of the citations in the manuscript and cross-checked them with the original sources to ensure that they are correct. We have also managed the citations using the Mendeley Reference Manager, which has helped us to keep track of all of the references and ensure that they are properly formatted according to the citation style guidelines.

We hope this version is satisfactory for highly respectful reviewer.

Reviewer 4 Report

Dear authors,

Thanks a lot for addressing all my comments that have been raised during my initial review. I am happy enough with your responses and clarifications made to my initial comments and to the modifications you have accomplished within the text.

Some additional Minor comments:

1.       Please check carefully once again the whole manuscript for minor spelling and grammar mistakes.

-       i.e. page 9, paragraph 4: You reported; “Vitamin C is works…”. Vitamin C is works or Vitamin
C works?

-        Page 11, 1st paragraph: Please check letters’ font and extra spaces  between words for compatibility.

-        etc.

Author Response

Dear authors,

Thanks a lot for addressing all my comments that have been raised during my initial review. I am happy enough with your responses and clarifications made to my initial comments and to the modifications you have accomplished within the text.

Authors’ response: We would like to express our sincere gratitude for taking the time to review our manuscript. Your comments were extremely helpful in guiding our revisions, and we appreciate the time and effort that you put into reviewing our work.

We have made changes to the manuscript based on your feedback, including clarifying certain sections, reorganizing the structure, and adding additional references where appropriate.

We believe that we have now addressed all of your concerns and that the manuscript is now ready for publication.

We hope that you find our revised version satisfactory, and we thank you once again for your valuable feedback.

Some additional Minor comments:

  1. Please check carefully once again the whole manuscript for minor spelling and grammar mistakes.

-       i.e. page 9, paragraph 4: You reported; “Vitamin C is works…”. Vitamin C is works or Vitamin
C works?

Authors’ response: Thank you for bringing to our attention the error in our manuscript on page 9, paragraph 4, where we wrote "Vitamin C is works...". We apologize for this mistake and have taken corrective action by revising the sentence to read "Vitamin C works..."

-        Page 11, 1st paragraph: Please check letters’ font and extra spaces  between words for compatibility.

Authors’ response: Thank you for your comment regarding the font and extra spaces between words on page 11, first paragraph of our manuscript. We appreciate your attention to detail and have taken corrective action to ensure that the letters' font and spacing are consistent and appropriate.

-        etc.

Authors’ response: We appreciate your careful reading of our manuscript and your attention to detail in identifying this error. Your feedback has helped us to improve the clarity and accuracy of our writing.

Reviewer 5 Report

Considering the changes that have been made to the present work, it can be considered to be an improved scientific contribution and there is a possibility that the work might contribute to its dissemination

Author Response

Considering the changes that have been made to the present work, it can be considered to be an improved scientific contribution and there is a possibility that the work might contribute to its dissemination.

Authors’ response: We would like to express my sincere gratitude for taking the time to review our manuscript.

Your insightful comments and suggestions have been invaluable in improving the quality of our work. With your assistance, we have been able to significantly enhance the manuscript.

We appreciate your careful reading of the manuscript and your thoughtful feedback.

Your comments have helped us to identify areas that needed further clarification, as well as to improve the overall structure and organization of the paper.

Once again, we are deeply grateful for your valuable input, and we believe that your comments have made a significant contribution to the quality of the final manuscript.

Thank you for your time and effort in reviewing our work. No action was needed based on this reviewer comments.